# Regulatory T cells confer a circadian signature on inflammatory arthritis

L. E. Hand[1], K. J. Gray[1], S. H. Dickson[1], D. A. Simpkins[1], D. W. Ray [ID] [2], J. E. Konkel[3], M. R. Hepworth [ID] [3] & J. E. Gibbs [ID] [1,3 ✉]

The circadian clock is an intrinsic oscillator that imparts 24 h rhythms on immunity. This clock drives rhythmic repression of inflammatory arthritis during the night in mice, but mechanisms underlying this effect are not clear. Here we show that the amplitude of intrinsic oscillators within macrophages and neutrophils is limited by the chronic inflammatory environment, suggesting that rhythms in inflammatory mediators might not be a direct consequence of intrinsic clocks. Anti-inflammatory regulatory T (Treg) cells within the joints show diurnal variation, with numbers peaking during the nadir of inflammation. Furthermore, the anti-inflammatory action of Treg cells on innate immune cells contributes to the night-time repression of inflammation. Treg cells do not seem to have intrinsic circadian oscillators, suggesting that rhythmic function might be a consequence of external signals. These data support a model in which non-rhythmic Treg cells are driven to rhythmic activity by systemic signals to confer a circadian signature to chronic arthritis.

[1] Centre for Biological Timing, Faculty of Biology, Medicine and Health, University of Manchester, Oxford Road, Manchester, UK. [2] NIHR Oxford Biomedical Research Centre, John Radcliffe Hospital, Oxford, UK and Oxford Centre for Diabetes, Endocrinology and Metabolism, University of Oxford, Oxford, UK. [3] Lydia Becker Institute of Immunology and Inflammation, University of Manchester, Oxford Road, Manchester, UK. ✉email: julie.gibbs@manchester.ac.uk

The circadian clock confers 24 h rhythms onto physiology. This allows plants and animals to align physiological processes with daily changes in the environment, such as lighting levels, temperature and food availability. In mammals, these 24 h biological rhythms are driven by cellular oscillators, a network of transcriptional–translational feedback loops. The immune system exhibits circadian variation in function[1]. This is in part due to cell-autonomous circadian clocks within immune cells, driving daily variation in cell function, including trafficking[2–4], cytokine release[5,6] and phagocytosis[7]. Additionally, rhythmic circadian outputs, such as hormones, can drive daily changes in immune cell function[8,9].

Human chronic inflammatory diseases, such as asthma and rheumatoid arthritis (RA) show diurnal variation in their symptoms and disease markers[10–14], yet little is known about circadian rhythms in the setting of chronic inflammatory disease. RA is a progressive inflammatory disease of the joints. This autoimmune disease manifests as a consequence of abnormal innate and adaptive immune responses, and a failure to resolve inflammation[15]. If not treated sufficiently, inflammatory arthritis causes joint destruction, tissue re-modelling and eventually disability.

Numerous different cell types within the affected joints produce and respond to pro-inflammatory signals, including resident cells (such as fibroblast-like synoviocytes (FLS), tissue-resident macrophages, chondrocytes and osteoclasts) and immune cells recruited to the joints (such as T lymphocytes, monocytes, macrophages and neutrophils). In addition to this mix of pro-inflammatory cells, regulatory T cells (Tregs) are found within arthritic joints[16]. Under healthy conditions, Tregs maintain immune tolerance, which is achieved through suppression of a plethora of inflammatory mediators via a number of well described mechanisms ranging from production of suppressive cytokines to contact-dependent action[17–19]. Loss of immune tolerance contributes to autoimmune diseases.

We recently showed, in a murine model of inflammatory arthritis (collagen-induced arthritis, CIA), robust diurnal variation in inflammatory markers within the inflamed joint, with a peak in pro-inflammatory and chemotactic cytokines during the light phase (rest phase for nocturnal rodents)[20]. This is in keeping with data from RA patients, where pro-inflammatory markers and joint stiffness peak during the early morning, at the end of the rest phase[14]. Now, we identify daily changes in the cellular environment of the inflamed joint and address circadian circuits operating within this site. We find that both resident and recruited immune cells within the joint show dampened circadian rhythmicity. Strikingly, numbers of Tregs exhibit diurnal variation, peaking in number during the dark (active) phase (the nadir of inflammation), despite having no clear overt intrinsic circadian rhythmicity. Together our data indicates that daily rhythms in chronic inflammatory disease may be a consequence of diurnal rhythms in Treg function, driven by external rhythmic cues.

## Results

**Immune cell clocks are dampened in chronic inflammation.** To explore the cellular make-up of inflamed joints at the peak and trough of inflammation[20], arthritic (CIA) and control mice were sacrificed at ZT6 (peak) or ZT18 (trough) and immune cells harvested from the inflamed limbs for flow cytometric analysis (gating strategy Supplementary Fig. 1). As before two distinct macrophage populations, defined by MHC II expression[21,22], were found in the joints (Fig. 1a). Numbers of both MHC II$^{low}$ and MHC II$^{high}$ macrophages significantly increased with CIA, but neither showed time-of-day differences within healthy or inflamed joints (Fig. 1b). To address whether the intrinsic

macrophage clock remains intact under chronic inflammation, QPCR was performed on sorted cells. Both macrophage populations from arthritic joints showed a striking loss of *Rev-erbα* gene expression at its (ZT6) peak, accompanied by impaired peak expression of *Per2* and *Cry1*, indicating broad disruption of the internal circadian clockwork (Fig. 1c). Arthritis induced significant neutrophil recruitment to the inflamed limbs, but again no time-of-day effect was seen (Fig. 1d, e). Analysis of clock genes within neutrophils also revealed loss of the effect of time-of-day on *Rev-erbα* expression (Fig. 1f).

**Treg cell numbers within arthritic joints has diurnal variation.** Whilst numbers of macrophages and neutrophils within inflamed joints did not show diurnal variation, quantification of CD3ε$^+$ cells (a pan T-cell marker) in the joints of arthritic mice surprisingly revealed significantly higher numbers at ZT18, the nadir of inflammation, compared to ZT6 (Fig. 2a). Analysis using a further panel of T cell markers revealed that it was CD4$^+$ (but not CD8$^+$) T cells that showed diurnal variation (Supplementary Fig. 2a–c). Of the CD4$^+$ populations analysed, Tregs were the most abundant, making up 40–60% of CD4$^+$ cells within inflamed joints (Supplementary Fig. 2a). Tregs showed dramatic day–night variation in numbers within inflamed joints whilst numbers of T$_h$1 and T$_h$17 cells remained consistent through time (Fig. 2b, c). Further analysis examined the activity and proliferation of Tregs within the inflamed joints at different phases of the day (Fig. 2d, e and Supplementary Fig. 3a). Analysis of Ki67 expression showed a time-of-day effect in naïve mice, with more recently proliferated cells at ZT18. Conversely EdU staining (which marks cells undergoing S-phase) did not show the same effect. EdU-labelling studies did reveal increased incorporation of EdU into Tregs from arthritic animals compared to naïve animals at ZT6. However, from both Ki67 and EdU staining we did not observe an increase in Tregs from inflamed mice at ZT18, suggesting that the increased Treg numbers at ZT18 was not a consequence of elevated Treg proliferation. Expression of neuropilin 1 (NRP1) which potentiates Treg function and survival, was heightened during the dark phase in arthritic mice. Statistical analysis revealed a significant interaction between time-of-day and disease, confirming that the effect of arthritis on NRP1 expression was greater at ZT18. However, expression of glucocorticoid-induced tumour necrosis factor receptor-related protein (GITR), which was significantly up-regulated by arthritis, showed no diurnal variation. In separate experiments, we examined the production of IL10 by Tregs from arthritic joints at ZT6 and ZT18 by ex vivo restimulation assays (Fig. 2f and Supplementary Fig. 3b). Total IL10$^+$ Treg numbers in the joints were increased at ZT18 compared to ZT6. By quantifying the percentage of Tregs present within the joints at these times which are IL10$^+$ we established that the capacity to secrete IL10 remains constant between time points. This indicates that it is the increase in cell numbers at ZT18, rather than a change in their individual suppressive activity, that drives daily variation in local inflammation. To address whether the increase in Treg numbers within the joint was a consequence of increased numbers in the circulation, peripheral blood samples were collected from arthritic mice (Fig. 2g). There was a trend for reduced circulating Tregs at ZT18. Quantification of Tregs in non-draining (inguinal) and draining (popliteal) lymph nodes of arthritic mice revealed no significant time-of-day difference (Fig. 2h). Taken together this suggests selective, time-of-day-dependent Treg accumulation in inflamed joints.

**Naïve Treg cells do not possess functional clock machinery.** Our data indicates that in the context of arthritis, the accumulation and

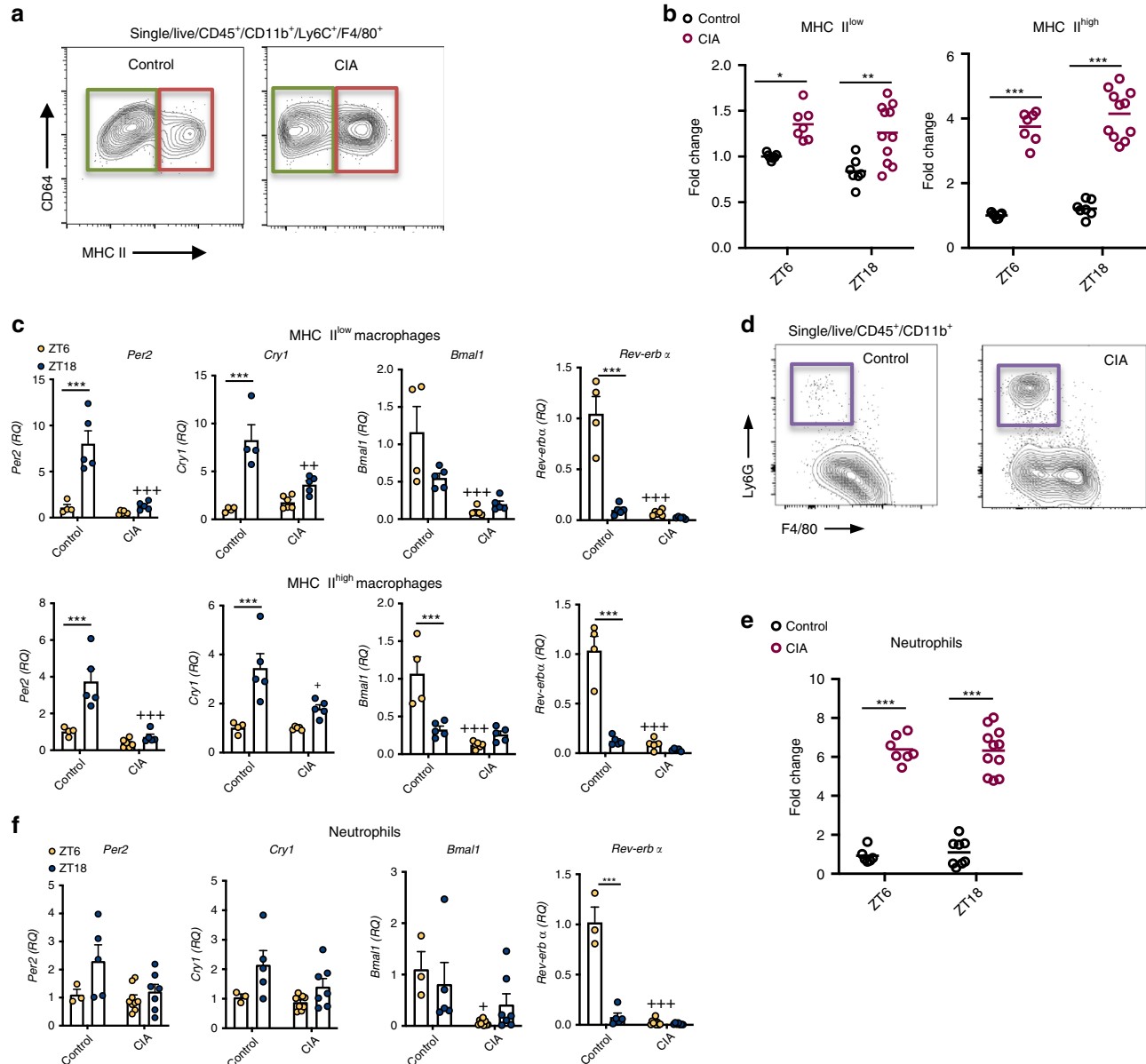

**Fig. 1 Macrophage and neutrophil rhythms under chronic inflammation. a** Two distinct macrophage populations were identified within joints, MHC II[low] (green box) and MHC II[high] (red box). **b** Numbers of MHC II[low] and MHCII[high] macrophages increased within the joints of arthritic animals, neither showed any time-of-day variation in numbers under control or arthritic conditions, data pooled from two separate experiments and normalised to control ZT6 mice (ZT6: control $n = 7$; CIA $n = 7$; ZT18: control $n = 7$; CIA $n = 11$), two-way ANOVA and Bonferonni post hoc tests. Graphs show individual data points with mean values. **c** MHC II[low] and MHC II[high] macrophages sorted from joints at ZT6 (control $n = 4$; CIA $n = 6$) or ZT18 (control $n = 5$; CIA $n = 5$) were analysed for clock gene expression (normalised to *gapdh*), data normalised to control ZT6, two-way ANOVA and post-hoc Bonferonni. Data are presented as mean values ± SEM. **d** Representative flow cytometry plots showing abundant neutrophil infiltration into the joints in CIA. **e** Numbers of neutrophils increased within the joints of arthritic animals, but did not show time-of-day variation in numbers under control or arthritic conditions, data pooled from two separate experiments and normalised to control ZT6 mice (ZT6: control $n = 6$; CIA $n = 9$; ZT18: control $n = 8$; CIA $n = 11$), two-way ANOVA and Bonferonni post hoc tests. Graph shows individual data points with mean values. **f** Clock gene expression (normalised to *gapdh*) in neutrophils sorted from joints at ZT6 (control $n = 3$; CIA $n = 9$) and ZT18 (control $n = 5$; CIA $n = 7$), data normalised to control ZT6, two-way ANOVA and post-hoc Bonferonni. Data are presented as mean values ± SEM. In all panels statistical significance between timepoints is shown as *$p < 0.05$, **$p < 0.01$ and ***$p < 0.005$ and significant statistical differences between treatment at given timepoints as +$p < 0.05$, ++$p < 0.01$ and +++$p < 0.005$. Source data are provided as a Source Data file.

function of Tregs could be regulated via circadian control. There are conflicting reports regarding the importance of the CD4[+] T cell intrinsic clock in regulating lymphocyte function[23,24]. Furthermore, the existence of an intrinsic clock in Tregs has not yet been addressed. To address this, we utilised mice in which *Bmal1* is targeted for deletion in T cells (PER2::luc CD4−*Bmal1*−/−) which prevents cell-autonomous circadian oscillations[6]. As

expected, CD4-*Bmal1*−/− mice showed significantly diminished expression of *Bmal1* within CD4[+] T cells, CD8[+] T cells and Tregs (but not CD4− dendritic cells) (Supplementary Fig. 4). Inguinal and popliteal lymph nodes from wildtype mice (PER2::luc *Bmal1*[flox/flox]) showed robust circadian rhythms (Fig. 3a). Deletion of *Bmal1* in T cells did not alter rhythmicity and there was no significant difference in circadian period between genotypes

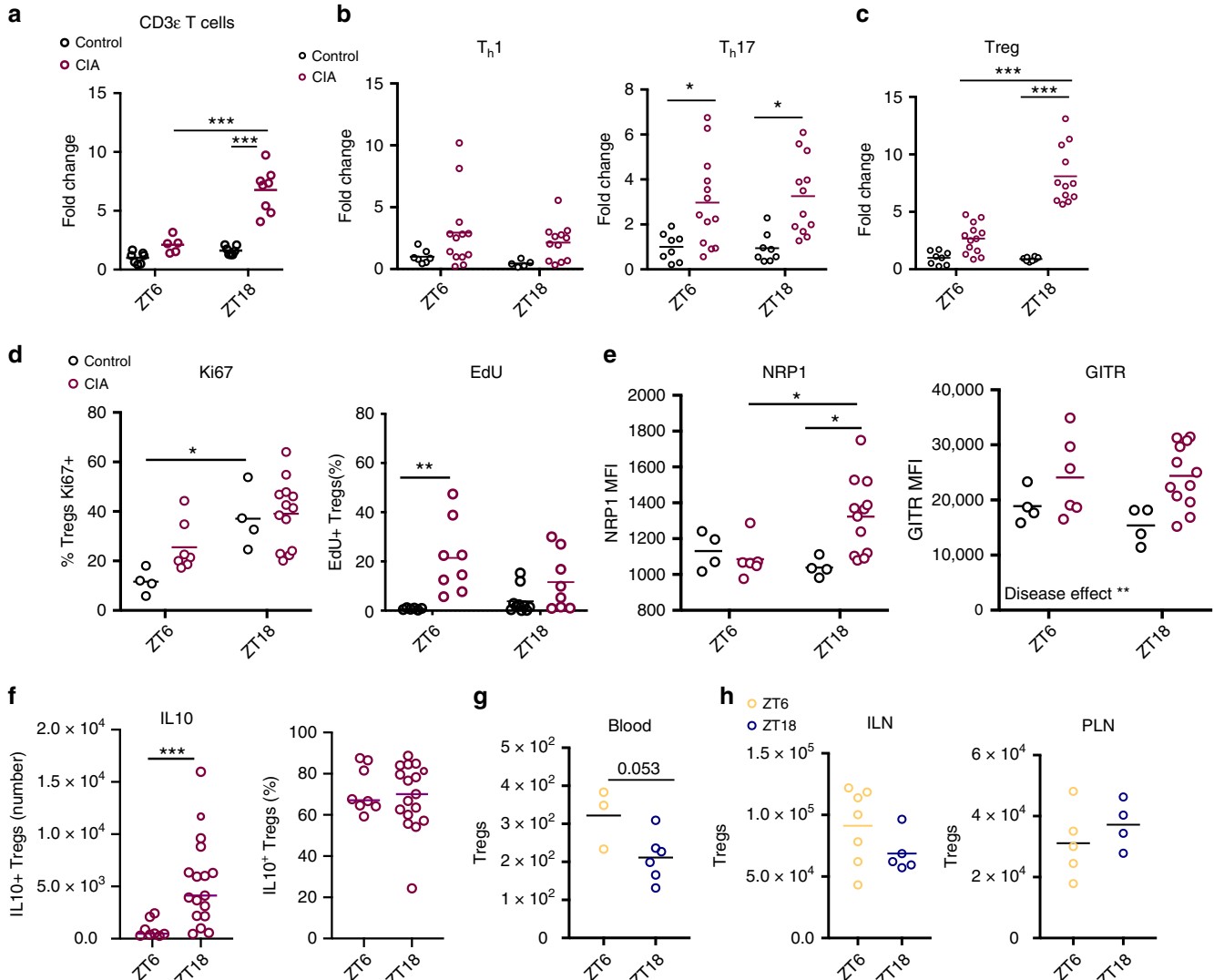

**Fig. 2 Diurnal variation in T cell populations within inflamed joints. a** CD3ε T cells (CD45+/CD11b−/CD3ε+) in limbs harvested from control and arthritic mice at ZT6 (control $n = 7$; CIA $n = 5$) and ZT18 (control $n = 7$; CIA, $n = 8$), data normalised to CD3ε T cell numbers in control limbs at ZT6, two-way ANOVA and Bonferonni post hoc tests. **b** Quantification of CD4+ T cell subsets (Th1 and Th17) in limbs at ZT6 (Th1: control $n = 7$; CIA $n = 13$; Th17: control $n = 8$; CIA $n = 13$) and ZT18 (Th1: control $n = 5$; CIA $n = 12$; Th17: control $n = 8$; CIA $n = 12$), data pooled from two separate experiments and normalised to ZT6 control animals, two-way ANOVA and post hoc Bonferonni. **c** Quantification of Tregs in limbs at ZT6 (control $n = 8$; CIA $n = 13$) and ZT18 (control $n = 8$; CIA $n = 12$), data pooled from two separate experiments and normalised to ZT6 control animals, two-way ANOVA and post hoc Bonferonni. **d** Expression of the proliferation markers Ki67 (ZT6: control $n = 4$; CIA $n = 7$; ZT18: control $n = 4$; CIA $n = 13$) and EdU (ZT6: control $n = 6$; CIA $n = 8$; ZT18: control $n = 10$; CIA $n = 8$) on Tregs isolated from control and inflamed joints, Kruskal–Wallis test and post-hoc Dunn's test. **e** Expression of NRP1 and GITR (ZT6: control $n = 4$; CIA $n = 6$; ZT18: control $n = 4$; CIA n $= 12$) on Tregs isolated from control and inflamed joints, two-way ANOVA and post-hoc Bonferonni test. NRP1—interaction between time-of-day and disease effect ($p = 0.0285$); GITR—disease effect ($p = 0.0077$). **f** IL10 expression by stimulated Tregs isolated from inflamed joints at ZT6 ($n = 8$) and ZT18 ($n = 17$) expressed as total number of IL10+ Tregs and % of Tregs that are IL10+, Mann–Whitney test. **g** Quantification of Tregs in blood of arthritic mice (ZT6 $n = 3$; ZT18 $n = 6$), two-sided unpaired T-test. **h** Quantification of Tregs in inguinal (ZT6 $n = 7$; ZT18 $n = 5$) and popliteal (ZT6 $n = 5$; ZT18 $n = 4$) lymph nodes from arthritic mice. Treg numbers were determined per lymph node, only popliteal lymph nodes draining inflamed limbs were analysed, two-sided unpaired T tests. All graphs show individual data points with mean values. In all panels statistical significance between indicated groups is shown as *$p < 0.05$, **$p < 0.01$ and ***$p < 0.005$. Source data are provided as a Source Data file.

(Fig. 3a). Splenic Tregs sorted from wildtype mice (PER2::luc $Bmal1^{flox/flox}$) at ZT6 and ZT18 showed negligible diurnal variation in expression of the core clock genes *Per2*, *Bmal1* and *Cry1*, but significant diurnal variation in *Rev-erbα*. Deletion of *Bmal1* caused down-regulation of *Rev-erbα* and up-regulation of *Cry1* as expected[25], but no effect on *Per2* (Fig. 3b). To characterise cellular circadian clock function with greater temporal resolution, Tregs were sorted from lymph nodes of naïve mice culled at 6 h intervals (Fig. 3c and Supplementary Fig. 5a, b). QPCR analysis revealed

that *Per2*, *Bmal1*, *Cry1* and *Per3* did not show rhythmicity. However, *Rev-erbα* did show significant differences in expression between time-points, peaking at ZT6. To confirm that antibody staining does not affect clock gene expression in Tregs, FoxP3^GFP cells were sorted from the lymph nodes of DEREG mice at ZT6 and ZT18 (with no prior antibody staining). Quantification of clock gene expression in these cells yielded concurrent results confirming lack of diurnal variation in all genes tested except *Rev-erbα* (Supplementary Fig. 5c, d). These data suggest that within

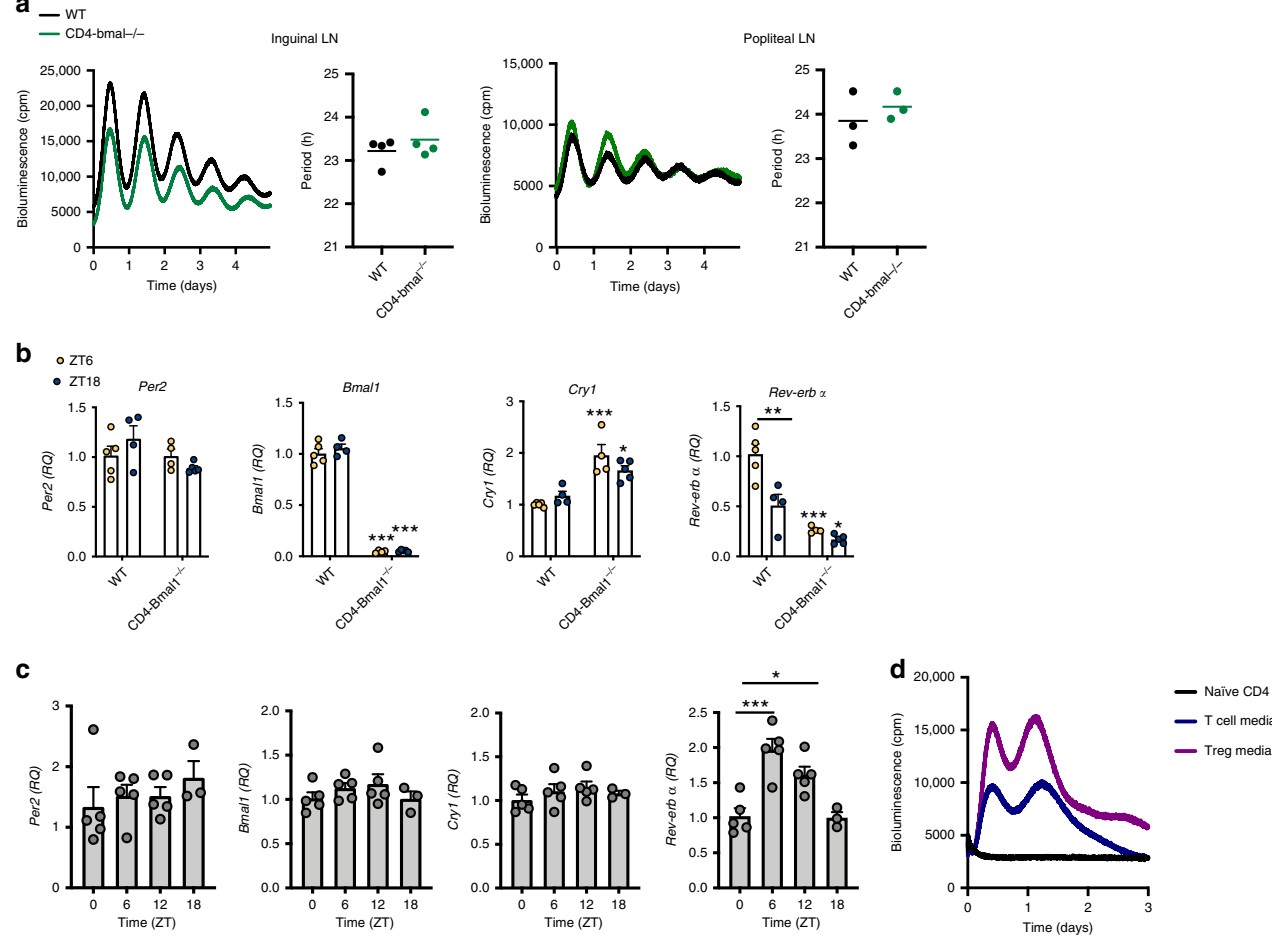

**Fig. 3 Treg cells do not have an intrinsic clock. a** Representative PMT traces and calculated period from paired inguinal ($n = 4$) and paired popliteal ($n = 3$) lymph nodes from CD4-Bmal1$^{-/-}$ and wildtype mice on a PER2::luc background. Graphs show individual data points and mean values. **b** Tregs were sorted from the spleens of wildtype and CD4-Bmal1$^{-/-}$ mice at ZT6 (WT $n = 5$; CD4-Bmal1$^{-/-}$ $n = 4$) and ZT18 (WT $n = 4$; CD4-Bmal1$^{-/-}$ $n = 5$) and utilised for QPCR to assess clock gene expression (normalised to *gapdh*), two-way ANOVA and post hoc Bonferonni. Unless indicated otherwise, significance shown compares genotype by time-of-day. Data are presented as mean values ± SEM. **c** Tregs were sorted from lymph nodes of mice at four time points across the day, identified by CD25$^{+}$CD127$^{low}$. Expression of clock genes (normalised to *gapdh*) was quantified by QPCR, normalised to ZT0 (ZT0,6,12 $n = 5$; ZT18 $n = 3$), one-way ANOVA and post hoc Bonferonni. Data are presented as mean values ± SEM. **d** PMT traces of naïve CD4$^{+}$ T cells purified from the lymph nodes of PER2::luc mice and cultured with IL2 alone, T cell media or Treg media, representative of three independent repeats. In all panels statistical significance is shown as $*p < 0.05$, $**p < 0.01$ and $***p < 0.005$. Source data are provided as a Source Data file.

lymph nodes and spleen, Tregs do not have a functional, autonomous circadian clock, but endogenous *Rev-erbα* gene expression retains circadian regulation, possibly in response to extrinsic signals as described before[24,26].

To determine whether Tregs become circadian synchronised during inflammatory challenge, naïve CD4$^{+}$ T cells were isolated from PER2::luc mice and cultured in either control media or conditions to promote T cell expansion (T cell media containing IL2) or Treg expansion (Treg media containing IL2, TGFβ and anti-IFNγ), both of which contained anti-mouse CD28 and anti-mouse CD3ε. By day 3 in Treg media, ~80% of CD4$^{+}$ cells were Tregs (Supplementary Fig. 6a). Stimulation (anti-CD3ε/anti-CD28) caused an induction of PER2 bioluminescence (Fig. 3d) indicating direct coupling of core clock gene expression to extrinsic T cell stimulation. A second peak was seen ~20 h later. It is unlikely that this second peak represents circadian activity as Tregs lacking *Bmal1* showed a similar PER2 induction after stimulation (Supplementary Fig. 6b, c). Instead this may be a consequence of the increase in cell numbers as they undergo proliferation in the expansion media.

**Glucocorticoids induce daily changes in Treg cell CXCR4.** Given that Tregs from inflamed joints show diurnal variation in activation markers, we tested whether naïve Tregs also show daily changes in phenotype. To this end we analysed expression of CXCR4, a chemokine receptor, on Tregs harvested from lymph nodes and spleen (Fig. 4a). CXCR4 expression showed time-of-day variation on naïve Tregs even in the absence of *Bmal1*, again supporting the critical role of an extrinsic signal conferring circadian control to aspects of Treg function (Fig. 4b). In contrast, the ligand for CXCR4 (CXCL12) did not show diurnal variation in circulating serum levels (Fig. 4c) Endogenous glucocorticoid levels, regulated by the circadian clock, are well-established to impose multiple effects on the immune system[27]. In vivo administration of dexamethasone, a synthetic glucocorticoid, at ZT0 (when endogenous glucocorticoid levels are low) induced a significant increase in CXCR4 expression in CD4$^{+}$ T cells, CD8$^{+}$ T cells and Tregs 4 h later (Fig. 4d). This supports the hypothesis that diurnal variation in CXCR4 expression in Tregs may be driven by a timed glucocorticoid signal. Serial blood sampling from naïve and arthritic mice

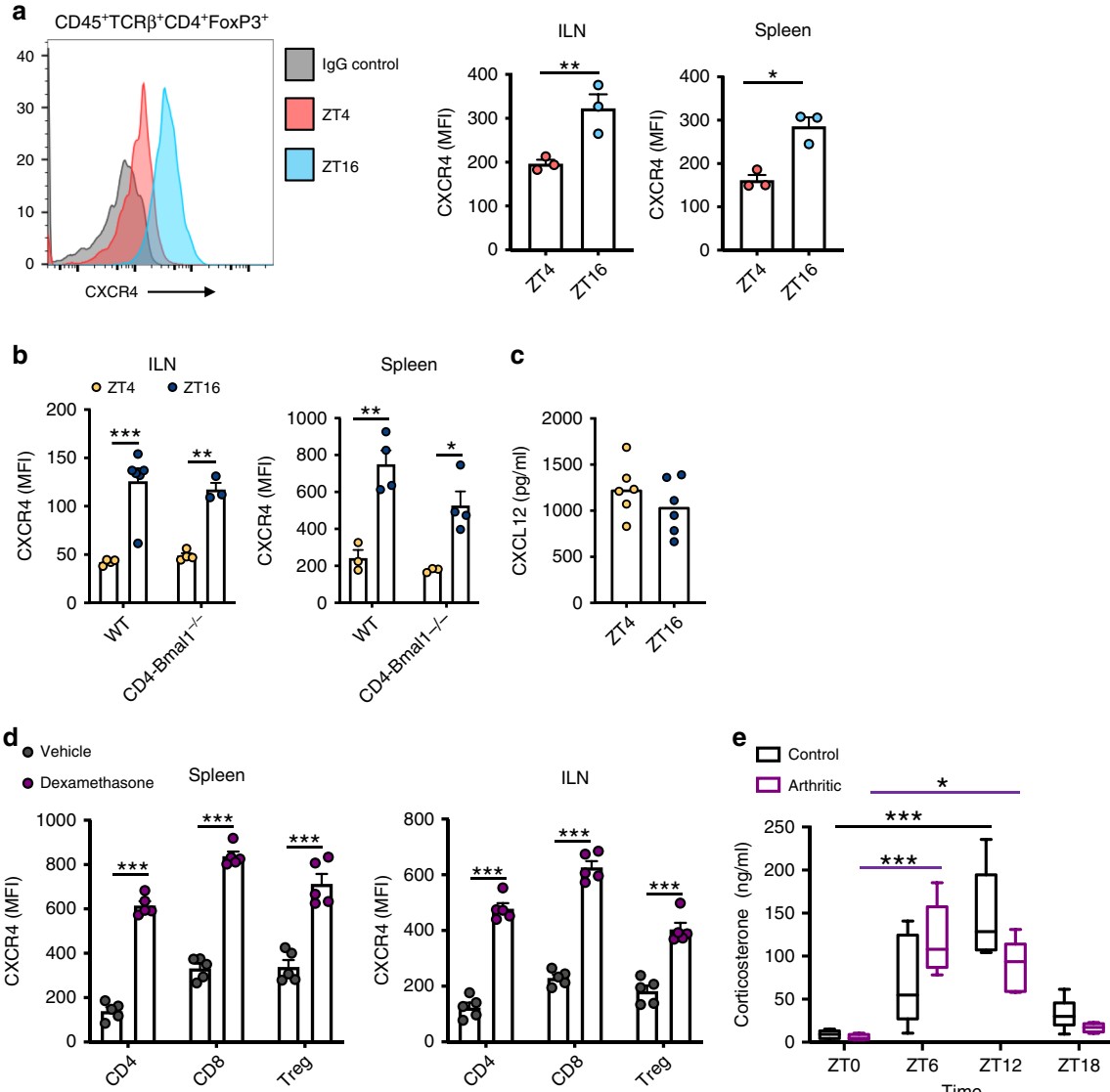

**Fig. 4 Glucocorticoid action on Treg cells. a** CXCR4 expression (geometric mean fluorescent intensity) on Tregs from spleen and inguinal lymph nodes of wildtype mice harvested at ZT4 (*n* = 3) or ZT16 (*n* = 3) two-sided unpaired *T*-test. Data are presented as mean values ± SEM. **b** Diurnal rhythmicity in CXCR4 expression persists in the absence of *Bmal1* ILN (ZT4: WT *n* = 4; CD4-Bmal1$^{-/-}$ *n* = 4; ZT16: WT *n* = 6; CD4-Bmal1$^{-/-}$ *n* = 3) and spleen (ZT4: WT *n* = 3; CD4-Bmal1$^{-/-}$ *n* = 3; ZT16: WT *n* = 4; CD4-Bmal1$^{-/-}$ *n* = 4), two-way ANOVA, post hoc Tukey. Data are presented as mean values ± SEM. **c** Plasma levels of CXCL12 in naïve mice at ZT4 and ZT16 (*n* = 6/group). Graph shows individual data points and mean values. **d** Effects of 4 h in vivo application of dexamethasone (2 mg/kg) or vehicle (cyclodextrin) at ZT0 on CXCR4 expression on the surface of T cell subsets derived from spleen or inguinal lymph nodes (*n* = 5/group). One-way ANOVA and post hoc Bonferonni. Data are presented as mean values ± SEM. **e** Plasma corticosterone levels in naïve (*n* = 6/time point) and arthritic (ZT0 *n* = 4; ZT6 *n* = 5; ZT12 *n* = 5; ZT18 *n* = 4) animals across a 24 h period (centre of box represents the medium and bounds extend from 25th to 75th percentile, whiskers are minima and maxima), two-way ANOVA (interaction between time-of-day and disease effect *p* = 0.0058; Time-of-day effect *p* < 0.0001; Disease effect NS) and post-hoc Tukey. In panels **a–d** statistical significance is shown as \**p* < 0.05, \*\**p* < 0.01 and \*\*\**p* < 0.005. In panel **e** \**p* < 0.05 and \*\*\**p* < 0.005 indicates significant statistical differences between the indicated time-points in naïve animals (black lines) and arthritic animals (purple lines). Source data are provided as a Source Data file.

revealed that diurnal variation in serum corticosterone levels persist in the setting of chronic inflammation, with peak concentration between ZT6 and ZT12 (Fig. 4e). Statistical analysis revealed a significant interaction between time-of-day and treatment, and a significant effect of time-of-day on corticosterone levels. Post hoc analysis revealed no significant differences between treatment groups at any time point, but significant increases above ZT0 values at ZT6 (arthritic only) and ZT12 (control and arthritic). This suggests the influence of rhythmic glucocorticoids on Treg function could persist even under chronic inflammatory conditions.

**Treg cell depletion enhances joint inflammation at night**. It is well established that Tregs play a protective role in the pathogenesis of arthritis. Adoptive transfer of antigen-specific Tregs impedes progression of CIA[28], whilst depletion of Tregs perpetuates disease progression, increasing disease score and paw swelling[29–31]. To test the hypothesis that the increase in abundance of these repressive cells at night confers time-of-day variation in disease markers we utilised DEREG mice, in which Tregs are depleted by administration of diphtheria toxin (DTX). We first confirmed our dosing paradigm (Fig. 5a) efficiently depleted Tregs, using lymph nodes as a test tissue to readout response.

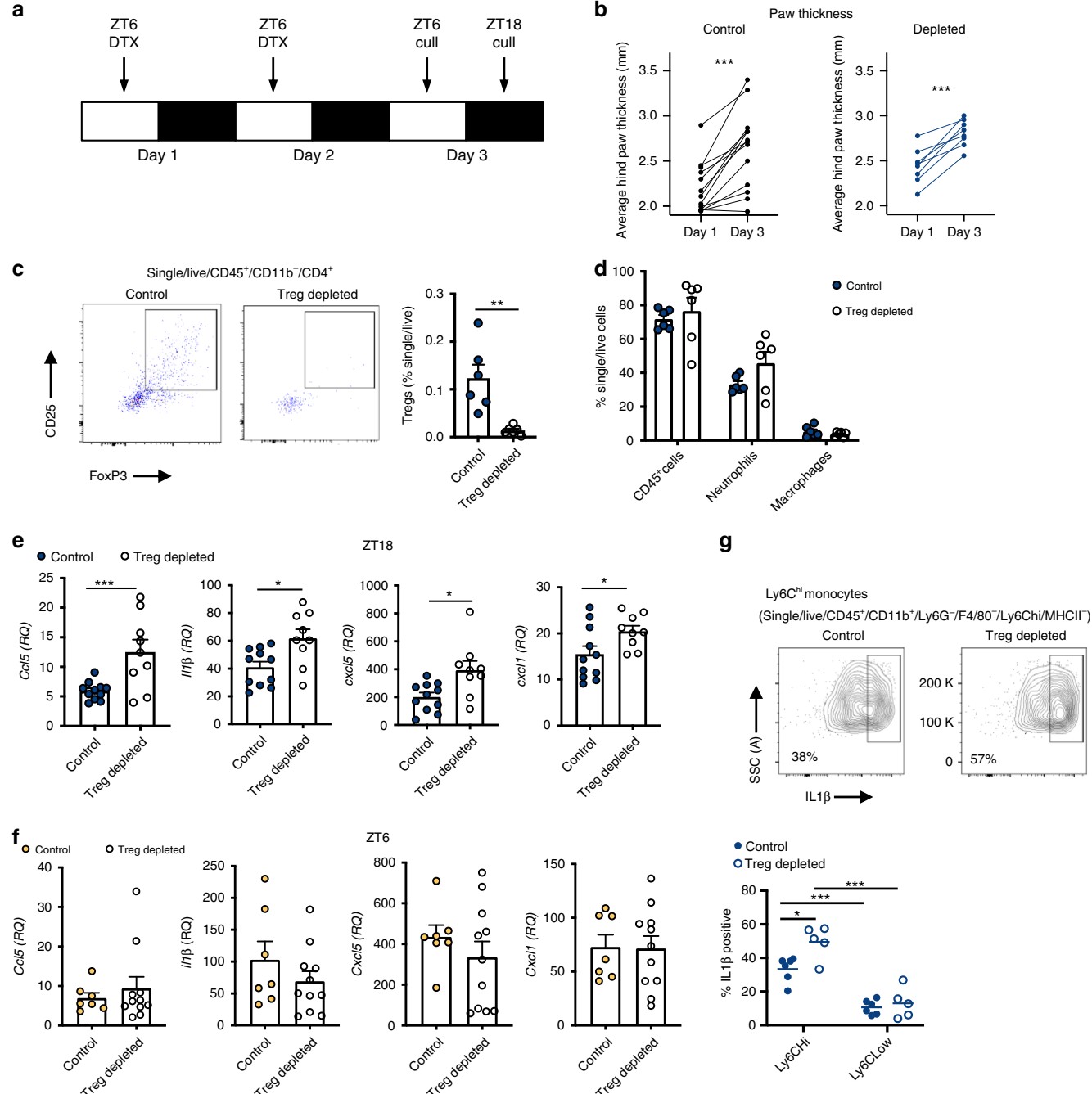

**Fig. 5 Depletion of Treg cells increases localised proinflammatory cytokine expression. a** Two daily injections of DTX at ZT6 led to significant depletion of Tregs 24–36 h after the last injection. **b** Progression of arthritis (as assessed by hind paw swelling) in control mice ($n = 13$) and in mice with depleted Tregs ($n = 8$), two-sided paired *T* tests. **c** Quantification of Tregs in arthritic control ($n = 6$) and Treg depleted ($n = 6$) animals, two-sided unpaired *T*-tests. Data are presented as mean values ± SEM. **d** Quantification of inflammatory cell populations within the inflamed joints in arthritic control ($n = 6$) and Treg-depleted mice ($n = 6$), one-way ANOVA. Data are presented as mean values ± SEM. **e** QPCR analysis of inflammatory genes (normalised to *gapdh*) within the joints of arthritic control ($n = 11$) and Treg-depleted ($n = 9$) mice culled at ZT18. Values were normalised to expression in control animals without arthritis, two-sided unpaired *t* tests. Data are presented as mean values ± SEM. **f** QPCR analysis of inflammatory genes (normalised to *gapdh*) within the joints of arthritic control ($n = 7$) and Treg-depleted ($n = 11$) mice culled at ZT6. Values were normalised to expression in control animals without arthritis. Data are presented as mean values ± SEM. **g** Effect of depletion of Tregs on IL-1β expression by joint monocytes (control $n = 6$; Treg depleted $n = 5$), two-way ANOVA and post hoc Bonferonni. Graph shows individual data points and mean values. In all panels statistical significance is shown as *$p < 0.05$, **$p < 0.01$ and ***$p < 0.005$. Source data are provided as a Source Data file.

Here we saw almost total (97%) depletion 24 and 36 h after the last dose (Supplementary Fig. 7a). DEREG mice (which are on a C57BL/6 background) were highly susceptible to arthritis (85% incidence), showing significant disease progression and paw swelling (Supplementary Fig. 7b). Furthermore, in keeping with our observations in DBA/1 mice[20], expression of proinflammatory cytokines in affected limbs was heightened at ZT6 versus ZT18, but this only reached statistical significance for

*Cxcl1* and *il6* (Supplementary Fig. 7c). In the first series of Treg depletion studies, DTX was administered once disease was established (observable paw swelling) and mice were culled 3 days later at ZT18 (the nadir of disease). During the treatment period, the disease continued to progress in both control and Treg-depleted animals (Fig. 5b). Flow cytometric analysis confirmed loss of Tregs within the inflamed joints after DTX treatment (Fig. 5c), but no significant alteration in numbers of neutrophils or macrophages (Fig. 5d). Analysis of 23 circulating serum cytokines revealed minimal effects of Treg depletion in the periphery, with only IL12p40 being up-regulated (Supplementary Table 1). However, analysis of the inflamed limbs showed that Treg depletion during established disease significantly increased expression of a subset of pro-inflammatory cytokines at ZT18 (*Ccl1, Il1β, Cxcl5* and *Cxcl1*) (Fig. 5e and Supplementary Fig. 7d). The effect of Treg depletion was diurnally regulated, in line with changes in Treg numbers, as a separate cohort of animals, culled at ZT6 (when Treg numbers are at their lowest) exhibited no change in expression of pro-inflammatory cytokines after DTX treatment (Fig. 5f). These data confirm a role for Tregs in conferring a time-of-day variation in arthritis severity.

**Treg cells repress proinflammatory monocyte responses**. The inhibitory effect of Tregs on T cells is well reported, but these anti-inflammatory cells also act on the innate immune system, suppressing the pro-inflammatory response of monocytes and macrophages[19,32]. To address which cells were responsible for the increased pro-inflammatory signal at ZT18 after depletion of Tregs, we repeated our depletion regime in DEREG mice and harvested cells from inflamed limbs at ZT18 to profile cytokine expression by different cell populations (gating strategy, Supplementary Fig. 8a). IL1β was chosen as a known rhythmic cytokine in CIA[20]. A robust IL1β signal was detected from monocytes and macrophages, and in Ly6C[hi] inflammatory monocytes this was significantly enhanced after depletion of Tregs (Fig. 5g and Supplementary Fig. 8b). Numbers of monocytes were not altered in the joints in the absence of Tregs (Supplementary Fig. 8c). Together this suggests that Tregs repress the pro-inflammatory function of Ly6C[hi] monocytes within the inflamed joint.

## Discussion

Chronic inflammatory arthritis is prevalent, and causes a large financial, personal, and social burden to society. Circadian variation in disease severity is frequent, but the mechanisms explaining this are undefined. Circadian disruption through genetic targeting of the core clock genes *Cry1/2* or *Bmal1* is associated with aggravated disease in the more simplistic mouse model of arthritis, collagen antibody-induced arthritis (CAIA)[22,33], providing evidence for the role of the clock in restraining the pathogenesis of this disease. Analysis of joint-derived macrophages and neutrophils revealed significant dampening of intrinsic clocks in these inflammatory cells under arthritic conditions. In keeping with in vitro studies, *Rev-erbα* expression was highly sensitive to inflammation[34]. Earlier work has identified changes in the expression of components of the core clock within synovial fibroblasts in the setting of chronic arthritis[35–38], but this is to our knowledge the first observation of the effects of chronic joint inflammation on the intrinsic clock within macrophages and neutrophils. Whether the amplitude of residual intrinsic oscillations in pro-inflammatory cells is sufficient to maintain circadian control of immune activities remains to be determined. However, we present a mechanism by which non-cell autonomous signals contribute to rhythmic repression in inflammation. We identify rhythmic accumulation of Tregs in inflamed joints, and show that Tregs within the tissue act on Ly6C[hi] monocytes to impair expression of inflammatory mediators. Together this suggests a mechanism whereby Tregs temporally gate pro-inflammatory processes. Despite the rhythmic behaviour of Tregs, they lack a functional, cell-autonomous circadian clock, instead responding to systemic timing cues, including the hypothalamic-pituitary-adrenal (HPA) axis.

We made a surprising observation that numbers of anti-inflammatory Tregs in the joints were significantly higher during the night, coincident with the observed nadir in inflammation. Other T cell populations, neutrophils and macrophages exhibited no temporal changes. Studies in pre-clinical models of arthritis show that enhancing Treg numbers, through adoptive transfer, improves disease outcome[28,39]. Thus, we predicted that the increased numbers of Tregs at night has a beneficial effect on disease. Not only were Treg numbers increased during the dark phase, but there was a trend for increased activity (NRP1 expression) at this time. However, Tregs harvested from joints at the peak and nadir of disease exhibited the same capacity to secrete IL10 upon stimulation, suggesting and it is the increase in numbers within the joints (rather than an increase in the suppressive activity of each cell) driving daily variation in local inflammation. Tregs isolated from arthritic joints at ZT18 did not show enhanced signs of recent proliferation, suggesting that increased numbers within the joints at night are more likely a consequence of recruitment. In keeping, expression of the chemokine receptor CXCR4 was increased on naïve Tregs during the dark-phase. These data suggest that Tregs are more pro-migratory during the dark-phase (a phenomenon that has been observed in naïve CD4[+] T cells[8]) and this may account for increased numbers.

Our observations that Tregs show phenotypic changes over 24 h, indicates that their function as immune regulators is under circadian control. Whilst CD8[+] T cells are established to exhibit robust low amplitude circadian rhythms[40]. Previous reports regarding whether CD4[+] T cells have an intrinsic clock are conflicting[23,24]. In agreement with others, we show that lymph nodes exhibit circadian rhythmicity in PER2::luc bioluminescence[2,41]. Deletion of *Bmal1* in T cells, which abolishes the clock, did not affect the lymph node clock, suggesting that T lymphocytes do not contribute significantly to the high amplitude rhythms in PER2 bioluminescence we observed from this tissue. Multiple other types of immune cells are found within the lymph nodes, some of which are known to be intrinsically rhythmic[5,42,43]. Furthermore, these cells are held within a capsule of connective tissue, which itself may exhibit rhythmicity. At a cellular level, weak oscillations in clock-dependent transcripts have been observed in both naïve human[23] and mouse[2] CD4[+] T cells. Notably *Rev-erbα* was found to oscillate in unstimulated cells from both species. Here, we find *Rev-erbα* mRNA varies through time in Tregs purified from lymph nodes, however other core clock genes (*Per2, Bmal1* and *Cry1*) intriguingly remain consistent throughout the day. This observation is supported by previous studies showing that *Rev-erbα* transcript in CD4[+] T cells is independent of *Bmal1* and likely regulated by extrinsic circadian factors; or zeitgebers[24]. Thus, we conclude that naïve Tregs do not have a synchronised cell-autonomous circadian clock. Furthermore, whilst stimulation of Tregs and CD4[+] T cells induces expression of PER2, this induction is insufficient to initiate and maintain a circadian oscillation, and is independent of *Bmal1*.

These data suggest that rhythmic extrinsic factors regulate Treg activity within the joint. These rhythmic signals may be locally derived in the joint or emanate from elsewhere, and could facilitate Treg recruitment to or retention within the joints. FLS play a key role in the pathogenesis of inflammatory arthritis[44], coordinating local inflammation through the secretion of cytokines

and chemokines. It is now established that FLS exist as multiple anatomically distinct subsets, each contributing differently to the inflammation and tissue damage associated with arthritis[45–47]. FLS are circadian rhythmic[22,35,38] and present as a candidate source for a locally derived rhythmic signal. Mechanisms driving Treg recruitment to arthritic joints are not well described. Tregs isolated from the synovial fluid of RA patients express higher levels of CCR4, CCR5 and CXCR4[44], which together have a multitude of chemokine ligands (including CCL17, CCL22, CCL16, CCL3, CCL3L1, CCL4, CCL5, CCL14 and CXCL12[48]), some of which are established to be produced by FLS[45]. The role of these rhythmic resident synovial fibroblast populations in directing rhythmic Treg biology warrants further investigation, but is beyond the scope of this study.

Candidates for systemic rhythmic extrinsic factors that regulate Treg activity include glucocorticoids. It is established that endogenous glucocorticoids drive diurnal oscillations in T cell function under healthy conditions[8,49,50]. Indeed, earlier studies have demonstrated that application of glucocorticoids can induce CXCR4 expression on CD4+ T cells and CD8+ T cells[8,49]. We now show that dexamethasone drives CXCR4 expression in Tregs, implicating activation of the glucocorticoid receptor as sufficient to impart a circadian signal. Whilst acute inflammatory events activate the HPA axis and enhance glucocorticoid secretion, this does not occur in inflammatory arthritis[51]. Instead, the amplitude and period of serum cortisol levels are similar to healthy controls[52]. In keeping, we observed maintenance of circadian rhythmicity in endogenous glucocorticoids in arthritic mice. Consequently, it is possible that in chronic inflammatory disease, endogenous rhythms in corticosterone are sufficient to drive diurnal variations in immune cell function.

Together, these data suggest that extrinsic circadian factors drive diurnal variation in Treg numbers within inflamed joints, leading to enhanced repression of inflammation during the night. We present data that rhythmic glucocorticoid signals direct daily changes in the phenotype of Tregs, however we acknowledge that other rhythmic endogenous signals may also contribute to diurnal regulation of Treg function. Localised depletion of Tregs does not affect numbers of inflammatory cells within the joints, but permits enhanced expression of pro-inflammatory genes. Tregs suppress a plethora of inflammatory mediators via a number of mechanisms, ranging from production of suppressive cytokines to contact-dependent action. They act not only on CD4+ effector T cells and CD8+ T cells but also dendritic cells and monocytes[17–19,53]. Our studies indicate that within the inflammatory milieu of the joint, Tregs repress expression of IL1β by Ly6Chi monocytes. This observation is supported by several in vitro studies[19,32,53] and a recent study in the setting of psoriasis where Tregs restrain the pro-inflammatory action of Ly6Chi cell populations in the skin[54]. The mechanistic nature of this interaction (cytokine mediated or contact dependent) is as yet unclear. Monocytes play a key role in arthritis, they are a source of pro-inflammatory cytokines, can polarise CD4+ T cells and can differentiate into osteoclasts and macrophages[55]. Peripheral blood monocytes can be divided into different classes based on their surface expression of Ly6C: Ly6Chi (classical), Ly6Cintermediate and Ly6Clo (non-classical). Typically, in autoimmune diseases classical monocytes are the predominant population causing tissue damage and disease progression[56]. In the context of murine models of arthritis, whilst classical monocytes are important[57], non-classical monocytes are also recruited to the joints where they drive tissue destruction[21,58]. Our data suggests that monocyte function is modulated in a time-of-day-dependent fashion by Tregs. It is highly likely that the cellular targets for the repressive action of Tregs extend beyond these myeloid cells, to include T effector cells, but we were unable to explore this with our assays.

Our findings provide a new mechanism by which the circadian clock regulates joint inflammation. We propose that rhythmic systemic circadian outputs, which are conserved in the presence of chronic inflammation, drive daily changes in the functional activity of anti-inflammatory Tregs, thereby imprinting diurnal variation onto arthritis expression via innate immune cells. There are conflicting reports regarding the suppressive capacity of Tregs isolated from the joints of RA patients, with some evidence of loss of functional regulation of immune tolerance[59,60]. However, a recent comprehensive profiling study of peripheral Tregs found no differences between RA and healthy individuals[61]. Our data suggests that time-of-day is an essential variable to consider in further clinical studies, and that the circuit we have identified supports the need for chronotherapy approaches to human RA.

## Methods

**Animals.** DBA/1 mice and C57BL/6 mice were purchased from ENVIGO (Huntingdon, UK) and Charles River (UK), respectively. Mice were generated (in house) to lack exon 8 of the *Bmal1* gene in T cells, by breeding Bmalflox/flox PER2::luc animals[6] with a CD4Cre/+ line[62]. The resulting offspring were either *Bmal1*flox/flox CD4Cre/+ (CD4-*Bmal1*−/−) or *Bmal1*flox/flox CD4+/+ (wildtype controls), all on a PER2::luc[63] background. DEpletion of REGulatory T cells (DEREG) mice[64], which carry a DTR-eGFP transgene under the control of a FoxP3 promoter, were kindly provided by Professor Tim Sparwasser (Universitats Medizin Mainz, Germany) and bred in house with local advice and guidance from Professor Mark Travis (University of Manchester). All experimental procedures were carried out under the UK Animals (Scientific Procedures) Act 1986 and were subject to local ethical review from the University of Manchester Animal Welfare and Ethical Review Board. All animals were housed in isolated ventilated cages under a 12:12 light:dark cycle, with ad libitum access to normal chow.

**Collagen-induced arthritis.** Male DBA/1 mice (9–10 weeks) were immunised (Day 0) with an intradermal injection (50 μL in two separate sites above the tail) of bovine type II collagen (mdbiosciences, Switzerland) emulsified in Complete Freunds's adjuvant (CFA, mdbiosciences) or as a control saline in CFA. A boost of bovine type II collagen was administered intraperitoneally on day 21. Arthritis was induced in male DEREG mice (8–15 weeks) by immunising with an intradermal injection (100 μL) of chicken type II collagen from sternal cartilage (Sigma, UK) emulsified in CFA on Day 0 and boosting with a second intradermal injection (100 μL) of chicken collagen and CFA on day 21[65]. Disease developed from approximately day 23 and animals were scored for disease severity on a scale of 0–4 per limb using the following scale: (1) one inflamed digit, (2) two or more inflamed digits, (3) swelling of the foot pad and minor ankylosis and (4) severe swelling of the foot pad and joint and severe ankylosis[20]. A score of 3 or 4 indicates significant inflammation involving the foot pad. Animals were sacrificed by cervical dislocation once one or more paws scored a 3 or more. Only limbs scoring 3 or 4 were utilised for downstream analysis. DEREG mice were administered DTX intraperitoneally (500 ng in 200 μL) to ablate Tregs. DTX was administered on 2 consecutive days during the mid-light phase (ZT6) before tissue collection the next day. For controls, either WT littermates were treated with DTX or animals expressing the DTX receptor were monitored but not treated with DTX. Limbs were dissected from naïve and arthritic mice 1 mm above the ankle or wrist joint and the skin was removed and tissue snap frozen.

**Flow cytometry and fluorescence-assisted cell sorting.** Cells were digested from limbs using 10 mg/mL collagenase from *Clostridium histolyticum* type IV (Sigma, UK)[20] with a digestion time of 45 min at 37 °C. Cells were extracted from lymph nodes and spleen using mechanical disruption. Cell numbers were quantified using an automated cell analyser, whereby cell suspensions were diluted 1:4 and stained with a solution of acridine orange and DAPI (Solution 18, Chemometec, Denmark). The cells were applied to an NC-Slide A8 which was analysed on the NucleoCounter NC250 (Chemometec) to provide total cell counts and cell viability. Cell staining was carried out in a 96-well V-bottomed plate. After live/dead staining (LIVE/DEAD fixable blue dead stain kit, ThermoFisher Scientific) and blocking (1:100 anti mouse CD16/CD32, Fisher Scientific, UK) cells were stained with a panel of antibodies (see Supplementary Table 2). All extracellular antibodies were utilised at 1:200 with the exception of biotin-conjugated CXCR4 which was utilised at a dilution of 1:100 with subsequent addition of streptavidin-PE (Invitrogen) at a final concentration of 0.8 μg/mL. For intracellular staining an eBioscience FoxP3/ Transcription factor staining kit (Thermo Fisher) was utilised as per kit instructions. Intracellular antibodies (FoxP3, Ki67, RORγt, TBET and IL10) were utilised at a 1:100 dilution and IL1β was utilised at 1:10 dilution. Where intracellular cytokine expression was analysed, isolated joint cells were first incubated in media containing monensin (1:1000, BioLegend UK) for 3.5 h at 37 °C before staining commenced. After application of antibodies targeting extracellular proteins, cells were incubated with a fixation/permeabilisation solution (FoxP3/Transcription

factor staining kit, Thermo Fisher) for 45 min before intracellular staining. Amine Reactive Compensation beads (Thermo Fisher Scientific) were utilised with the LIVE/DEAD stain for compensation controls. One Comp eBeads (Thermo Fisher Scientific) were utilised to produce single-stained compensation controls. Cells were analysed on an LSR II or BD Biosciences LSR Fortessa and data analysed using FlowJo. Fluorescence-assisted cell sorting was undertaken using either a BD Biosciences FACS Aria Fusion or BD Influx (BD Biosciences).

**Flow cytometry on blood**. Blood was collected 1:1 into 3.9% sodium citrate from the tail vein of control or arthritic mice. Cells were stained with a panel of antibodies (see Supplementary Table 2) for 25 min and red blood cell lysis was performed (RBC lysis buffer, Thermo Fisher). Cells were then processed as described above.

**T cell stimulation assay**. Cells were digested from the limbs of arthritic mice harvested at ZT6 or ZT18. Cells were counted (BioRad TC20 automated cell counter) in the presence of trypan blue. Samples were washed and re-suspended in stimulation cocktail (containing 0.1 μg/mL ionomycin, 12.5 μg/mL PMA and 1:1000 monesin solution (Biolegend)) for 3.5 h at 37 °C. After washing, cells were utilised for flow cytometry as described above, where intracellular staining (FoxP3 and IL10) was performed utilising the FoxP3/Transcription factor staining kit (Thermo Fisher).

**EdU staining**. EdU staining was performed using the Click-iT Plus EdU Alexa Fluor 647 Flow cytometry assay kit (Life Technologies). 4 h prior to harvest, mice were administered EdU (2.5 mg/mL, 200 μL intraperitoneally). Cells were digested from the limbs as described and stained with live/dead solution, and cell surface antibodies as described above. Samples were then washed, and fixed with 50 μL Click-iT fixative, before washing and re-suspending in PBS overnight at 4 °C. The next day, samples were washed (PBS with 1% BSA) and re-suspended in Click-iT reaction cocktail for 30 min before washing (Click-iT saponin-based permeabilization and wash reagent) and staining with intracellular antibodies. After a final wash, cells were re-suspended in FACS buffer for analysis. Cells were analysed on a BD Biosciences LSR Fortessa and data analysed using FlowJo.

**Sorting Treg cells from spleen and lymph nodes**. Cells were isolated from the spleen and lymph nodes via mechanical dissociation. After blocking (1:100 anti-mouse CD16/CD32, Fisher Scientific, UK), cells were stained with a panel of extracellular cytokines and DAPI (0.25 μg/mL) added to allow discrimination of dead cells. Tregs were identified based on expression of CD45$^+$ CD3ε$^+$ CD4$^+$, CD25$^+$ CD127$^{low}$. Cells (~50,000) were sorted directly into RLT buffer (RNeasy plus micro kit, Qiagen) and stored at −80 °C until RNA extraction using standard protocols.

**Photomultiplier Tube analysis**. Lymph nodes from mice expressing the PER2::luc reporter were first synchronised with dexamethasome (200 μM) before being placed onto Millicell cell culture inserts within 35 mm dishes containing recording media[66] and sealed with a glass coverslip. For recordings from T cells, lymphoid cells were dissociated from lymph nodes (mesenteric and inguinal) via mechanical disruption through a 40 μM filter. Naïve CD4$^+$ T cells were purified from the mixture (EasySep mouse naïve CD4$^+$ T cell isolation kit, Stemcell Technologies). Purified CD4$^+$ T cells were then cultured in "T cell recording media" (supplemented with non-essential amino acids and 50 nM β-mercaptoethanol) containing anti-mouse CD28 (1 μg/mL) and IL2 (10 ng/mL) on cell culture plates coated with anti-mouse CD3ε (3 μg/mL). To drive naïve CD4$^+$ T cells to differentiate into Tregs, mouse recombinant TGFβ (2 ng/mL) and anti-IFNγ antibody (10 μg/mL) were added to the media ("Treg media"). For re-stimulation cells were removed, centrifuged and re-suspended in media before re-plating onto anti-mouse CD3ε-coated plates and re-stimulated with CD28. Data was captured and period analysed using MFourFit in BioDare 2 (www.biodare2.ed.ac.uk).

**QPCR assays**. QPCR was performed on RNA isolated from cells and tissues using RNeasy kits (mini or micro) or Trizol extraction methods. Frozen limbs were ground up in liquid nitrogen prior to Trizol RNA extraction. RNA was converted to cDNA and QPCR performed using TaqMan QPCR primers and probes (Supplementary Table 3) using the StepOne System (Thermo Fisher). β-actin or gapdh were utilised as housekeeping genes.

**Corticosterone ELISA**. Serial samples of blood were collected from the tail vein of arthritic and naïve DBA/1 mice at four timepoints over 24 h. These were centrifuged (10,000 × g for 10 min at 4 °C) to produce serum which was stored at −80 °C until the assay was performed. The corticosterone ELISA was performed using the small volume protocol with 10 μL serum as per kit instructions (Enzo Life Sciences, UK).

**CXCL12 ELISA**. Terminal blood samples were collected from naïve mice at ZT4 and ZT16. These were centrifuged (10,000 × g for 10 min at 4 °C) to produce

serum, which was stored at −80 °C until the assay was performed. The CXCL12 ELISA was performed as per kit instructions using a DuoSet ELISA (R&D Systems).

**Bio-plex assays**. Serum harvested from arthritic DEREG mice was stored at −80 °C until cytokine analysis. A Bio-Plex Pro Mouse cytokine 23-plex assay was utilized on a Bio-Plex 200 system (Bio-Rad, UK) as per kit instructions, with samples prepared 1:4 in sample diluent.

**Glucocorticoid injections**. Water-soluble dexamethasone (dexamathasone–cyclodextrin complex, D2915, Sigma) was dissolved in saline (400 μg/mL). Mice received 2 mg/kg dexamethasone or vehicle (cyclodextrin) in an intraperitoneal injection at ZT0. 4 h later, mice were sacrificed and the inguinal lymph nodes and spleen harvested. Cell were extracted and stained with antibodies against extracellular markers (including a biotin-labelled CXCR4 antibody). After washing, cells were incubated with streptavin-PE (1:250) before fixation and permeabilization (FoxP3/transcription factor-staining kit) and intracellular staining for FoxP3.

**Statistical analysis and reproducibility**. Flow cytometry data was analysed using Flow Jo version 10.5. Statistical analysis was performed using GraphPad Prism version 7.0. Unless stated otherwise, data is presented as mean ± standard error of the mean (SEM). Comparisons between two groups were made using two-tailed Student's t tests. Comparisons between multiple groups were made using one-way ANOVAs and appropriate post-hoc tests with correction for multiple comparisons as defined in the figures. Where two variables (e.g. time-of-day and treatment) were included, two-Way ANOVAs were utilised with appropriate post-hoc tests with correction for multiple comparisons as defined in the figures. p values for interaction and main effects are shown on Supplementary Table 4. Statistical significance is taken as *$p < 0.05$, **$p < 0.01$ and ***$p < 0.005$ and +$p < 0.05$, ++$p < 0.01$ and +++$p < 0.005$. All in vitro studies were successfully repeated on three separate occasions with similar results. In vivo experiments were replicated with similar results at least twice during independent experiments under identical conditions.

**Reporting summary**. Further information on research design is available in the Nature Research Reporting Summary linked to this article.

## Data availability

The authors declare that the data supporting the findings of this study are available within the paper and its supplementary information or from the corresponding author upon reasonable request. The source data underlying all figures and Supplementary figures are provided in the Source Data file.

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

## Acknowledgements

We would like to thank the staff of the University of Manchester Flow cytometry facilities, Dr. Gareth Howell and Mr. Michael Jackson. We are grateful to Professor Tim Sparwasser (Universitats Medizin Mainz, Germany) for the provision of the DEREG mice and Professor Mark Travis (University of Manchester) for advice on breeding and

use of this strain. This study was funded by an MRC grant (MR/L018640/1) and a Versus Arthritis Career Development Fellowship awarded to Julie Gibbs (20629) and an MRC grant (MR/S002715/1) awarded to Julie Gibbs, Joanne Konkel and David Ray. Research in the Hepworth lab is supported by a Sir Henry Dale Fellowship jointly funded by the Wellcome Trust and the Royal Society (Grant no. 105644/Z/14/Z) and a Lister Institute Research Prize. Research in the Ray lab is supported by an MRC programme grant (MR/P023576/1). D.W.R. is a Wellcome Investigator, Wellcome Trust (107849/Z/15/Z).

## Author contributions

L.E.H. contributed to the experimental design, performed some of the research and associated data analysis. K.J.G. contributed to the experimental design, performed some of the research and associated data analysis. S.H.D. performed some of the research and associated data analysis. D.A.S. performed some of the research. D.W.R. contributed to the production of the manuscript. J.E.K. contributed to the experimental design and production of the manuscript. M.R.H. contributed to the experimental design and production of the manuscript. J.E.G. contributed to the experimental design, performed some of the research, contributed to the data analysis, supervised the work and production of the manuscript.

## Competing interests

The authors declare no competing interests.
