## [Peer Review File · Nature Communications]

Reviewers' comments:

Reviewer #1 (Remarks to the Author):

In this interesting manuscript, Hand et al present a follow-up of their 2016 paper which had shown a day-night difference in the outcome of experimental arthritis (collagen-induced arthritis) in mice. The new manuscript aims at finding the reason why inflammation in this model is higher when collagen injection is done in the day. The authors identify regulatory T cells (Tregs) as a key player in this rhythm, with higher function of these cells at night, which represses more the monocyte-mediated inflammation than in the day. They further show that the circadian regulation is not intrinsic to Tregs. This is a nice study, with an important question, and well-performed work. It is also, to my knowledge, the first study to address directly the question of a clock in Treg, and one of the first to address the circadian regulation of Tregs.

Main concerns:

1) The experiments about the Treg clock present some issues:

- The CD4-Cre driver does not restrict gene deletion to CD4 T cells, as is mentioned in many places in the manuscript. Since Cre is expressed in all CD4-expressing cells, including in CD4+CD8+ thymocytes, all mature T cells (both CD4+ and CD8+) will lack Bmal1 in their model. The whole manuscript should be revised accordingly, including the interpretation of the data.
- A better characterization of the KOs should be performed, looking at Bmal1 expression not only in Tregs, but also in other cell types.
- The authors use the remaining rhythms in LNs of T cell Bmal1 KOs to support their conclusion that there is no T cell clock (or no robust T cell clock). However, it may just be that there are still cells in the LNs that have high amplitude rhythms, masking any lack of rhythms in the T cells. This is discussed in the Discussion, but one cannot use these data as evidence for their point on page 7 (i.e. absence of clock in Tregs). Another reason for which there might be little effect of the T cell Bmal1 KO is that Per2 is well known (at least in the liver) to be responsive to systemic signals and thus, to be rhythmic even in the absence of a local clock (e.g. by abrogating Bmal1 expression).
- About the experiments looking at clock gene expression by PCR on Treg RNA (Fig 3b, c), have the authors considered that the sorting procedure might have impacted on clock gene expression? And in the case of Fig 3b, I would be more cautious with interpretation of the effects of the KO, as only 2 time points were used.

2) Based on the previous point (absence of Treg-intrinsic clock regulation), the authors address possible external timing cues. Based on previous circadian T cell literature (which the authors cite), glucocorticoids and CXCR4 regulation were obvious candidates to test. The authors do this by looking at CXCR4 expression on Tregs, and then by exogenous administration of dexamethasone. These are nice experiments and the results are interesting. However, although they are consistent with the model, these data do not prove that this is the only or even the main mechanism for circadian regulation of Treg in their model, as stated in different places in the manuscript (e.g., line 269-270, line 314).

3) One very interesting aspect of the model put forward by the authors is that the Tregs confer rhythmicity in the CIA model via an effect on myeloid cells, in particular the Ly6Chi monocytes. Unfortunately, this part of the study lacks firm demonstration. The authors write (line 225-226): "These data confirm a role for Tregs in conferring a time of day variation in arthritis severity." However, the report is lacking experiments specifically comparing disease progression or paw inflammation after CIA at ZT6 vs 18, in the context of Treg depletion. Such data would strengthen the manuscript. Also, the link between the Treg rhythms and the monocytes is based only on the impact of Treg depletion on monocyte counts and IL1beta expression, but the effect could be indirect. Moreover, the authors wrote in the Discussion (line 342-344): "It is highly likely that the cellular targets for the repressive action of Tregs extend beyond these myeloid cells, to include T effector cells, but we were unable to explore this with our assays". Why do they think that the

picture is incomplete? Why could this question not be explored? These doubts expressed by the authors support my concerns raised above, that the Treg-myeloid connection might be either indirect or only a part of the mechanism. In addition to this, how the synoviocytes (and their clock) affect or mediate these processes (including the involvement of Tregs) is unclear and should be addressed.

4) It would be interesting that the Discussion address the results in the context of other literature on rodent experimental models of arthritis. The authors cite their previous studies (Hand et al, 2016, 2019), but it is surprising that there is no discussion at all in the context of the Hand et al 2019 paper, which showed a role of the clock in mesenchymal cells to control the disease. This is particularly interesting in the context of the current manuscript, which suggests that the Treg circadian regulation is not relying on a clock within Tregs. Studies by Hashiramoto et al, JI, 2010, and Yoshida et al, Scand J Rheumatol, 2013, should also be discussed.

More specific concerns:

5) Conclusions about a dampening of the macrophage and neutrophil clocks in the inflamed joints are a bit overstated given that the data are only for two time points. Moreover, in the case of the neutrophils, although *Reverba* expression is reduced, the other clock genes seem to retain a time-dependent variation.

6) It is good that the FACS gating strategy is shown for some stainings, but this should be the case for all stainings. Examples of FACS plots and gating strategy should be shown for all stainings, including CD4/CD8 T cells (suppl. fig. 2), Treg markers (Fig 2d).

7) About *NRP1* expression (Fig 2d), it is said that it is higher in the night, but the figure seems to show a non significant difference, although the lack of significance could be due to a lack of statistical power, and an increase in the group size might make it significant.

8) Please check the whole manuscript for the format of gene/transcript names, and remain consistent (capital letter or not; mouse genes should have first letter cap).

9) Fig 4b has a problem with the indication of stats: the difference is between time points, not genotypes.

10) Regarding Fig. 4d, it is said (line 194) that arthritic mice are "similar to controls". This is vague. What do the stats say? Is there an interaction in the ANOVA? An effect of genotype?

11) Why are some graphs with clouds of dots (one dot per sample) while other graphs are bar graphs? Clouds of dots give a more direct indication of the actual data and the variability, and should be preferred, I think.

12) About suppl. fig 5c, it is said that pro-inflammatory cytokines are more expressed at ZT6 than at ZT18. The stats do not support this for 4 out of 6 cytokines/chemokines. The group size should be increased to have more statistical power?

13) Line 213 says that the disease progresses both in control and Treg-depleted mice, but Fig 5b does not show a statistically significant progression.

14) In Suppl Fig 6a, are the F4/80⁻ cells really negative? How is the gate set?

15) Fig 5g: A 2-way ANOVA should be performed, not 1-way.

16) It is good that the authors have provided a table of the antibodies used in the study, but the

fluorophores and suppliers should be added.

17) For the QPCRs, the TaqMan probe sets which were used should be listed. How the two control genes were used should also be described (average of Cts?).

Reviewer #2 (Remarks to the Author):

The authors make the novel observation that Tregs, as a critical immune cell population, observe time-of-day differences in inflamed arthritic joints. The authors then go on to show that absence of Tregs ablates time-of-day rhythmicity in inflammatory parameters. While the findings is very interesting and generally well demonstrated, the explanation of altered Treg numbers and function in joints should be developed more.

Main points:

Suppl. Fig. 2 CD4 and Fig. 2b and C: There is in the CIA condition a big increase in the CD4 compartment at ZT18 (3 fold higher compared to ZT6) yet, Th1 and Th17 are not affected. The only increase is seen in Tregs. Are the authors saying that these are the predominant populations of CD4 T cells in this scenario? It would be important then to show absolute numbers of the total and individual CD4 populations to support that claim.

Fig. 2d: With respect to the proliferation of Ki67+ cells, since the time-of-day-difference is stronger in untreated mice and the level of Ki67+ cells is comparable between untreated and arthritic mice at ZT18, why are Treg numbers not increased at ZT18 in the steady state? Since the etiology of higher Treg numbers is important for the paper the authors should also consider performing additional experiments with BrdU/Edu to verify that Tregs have indeed proliferated. Are the cells recruited from blood or do they proliferate locally? Page 6, line 124-125: '... as expression at ZT6 was enhanced by inflammation.' Is this the case, i.e. is expression of Ki67 enhanced on a per cell basis or are the more cells positive for Ki67?

Additionally, it would be important to show that activity of Tregs is heightened at specific times, e.g. using a functional assay, not just a surface staining and/or they should investigate the levels of IL-10 in the joints. NRP1 staining is not significant, at least this is not indicated. Are the stats mislabeled between NRP1 and GITR? These molecules should be explained.

As the major inflammatory cell types in the joint (presumably neutrophils and monocytes/macrophages, the absolute numbers of CD45 subpopulations should be shown to be able to discuss this) show no time-of-day difference, what do the authors propose is the cell type that mediates the rhythmic changes with respect to inflammatory swelling and paw thickness? Are the authors saying that it is solely regulated at the anti-inflammatory level by Tregs? DEREK (+DTX) mice still show oscillations in Cxcl1 expression, indicating pro-inflammatory cells to be involved as well. This should be discussed more.

Since this is the first time that Tregs have been investigated for circadian clock gene expression, and since this is important for the conclusion of the story, the authors should include possibly a few more clock genes. E.g., from the studies cited, it seems that Per3 was the gene, together with Rev-Erba that oscillated the most.

Fig. 5e-f: It would be important to show the data from control and DEREK-DTX mice in the same graph to assess whether oscillations are ablated. These seem indeed to be ablated, with the exception of Cxcl1.

What is the (rhythmic?) phenotype of the Cd4Bmal1 KO mice/Tregs in the joints with respect to

arthritis? Since the authors have previously published with the Reverba KO mouse and since Reverba is the only rhythmic clock gene: Do these KO mice have rhythms in arthritis?

Minor:

Fig 1C: statistics should be shown between control and inflamed groups as all clock genes, including Bmal1, are dramatically reduced, which should be noted.

What is the gating difference between Suppl Fig. 1 and 6 with respect to inflammatory monocytes? Are the authors missing inflammatory monocytes in Suppl Fig. 1?

Fig. Suppl. 2: The fold change normalized to at ZT6 should be 1 not 10.

What tissues do 'limbs' include, muscle of the leg? Or is it only the joints?

Fig. 2e: The y axis should be labeled.

Response to Reviewers' comments

Reviewer #1 (Remarks to the author)

In this interesting manuscript, Hand *et al* present a follow-up of their 2016 paper which had shown a day-night difference in the outcome of experimental arthritis (collagen-induced arthritis) in mice. The new manuscript aims at finding the reason why inflammation in this model is higher when collagen injection is done in the day. The authors identify regulatory T cells (Tregs) as a key player in this rhythm, with higher function of these cells at night, which represses more the monocyte-mediated inflammation than in the day. They further show that the circadian regulation is not intrinsic to Tregs. This is a nice study, with an important question, and well-performed work. It is also, to my knowledge, the first study to address directly the question of a clock in Treg, and one of the first to address the circadian regulation of Tregs.

Main concerns:

1. The experiments about the Treg clock present some issues:

1.1 The CD4-Cre driver does not restrict gene deletion to CD4 T cells, as is mentioned in many places in the manuscript. Since Cre is expressed in all CD4-expressing cells, including in CD4⁺CD8⁺ thymocytes, all mature T cells (both CD4⁺ and CD8⁺) will lack *Bmal1* in their model. The whole manuscript should be revised accordingly, including the interpretation of the data. A better characterization of the KOs should be performed, looking at *Bmal1* expression not only in Tregs, but also in other cell types.

We thank the reviewer for this correction, and have amended our manuscript accordingly to reflect the fact that the CD4^{Cre} driver deletes *Bmal1* in all T cells, rather than CD4⁺ T cells. We have performed additional analysis of *Bmal1* expression in Tregs, CD4⁺ T cells (excluding Tregs), CD8⁺ T cells and dendritic cells sorted from the spleen of these mice. As the reviewer predicted, *Bmal1* expression was significantly diminished in Tregs, CD4⁺ T cells and CD8⁺ T cells isolated from CD4-Bmal1^{-/-} animals. We utilised dendritic cells as a control population (those that do not express CD4) to show that *Bmal1* expression is not affected in these cells. We agree that it was important to better characterise the CD4-Bmal1^{-/-} animals and have now included this data in **Supplementary Figure 4**.

1.2 The authors use the remaining rhythms in LNs of T cell *Bmal1* KOs to support their conclusion that there is no T cell clock (or no robust T cell clock). However, it may just be that there are still cells in the LNs that have high amplitude rhythms, masking any lack of rhythms in the T cells. This is discussed in the Discussion, but one cannot use these data as evidence for their point on page 7 (i.e. absence of clock in Tregs). Another reason for which there might be little effect of the T cell *Bmal1* KO is that *Per2* is well known (at least in the liver) to be responsive to systemic signals and thus, to be rhythmic even in the absence of a local clock (e.g. by abrogating *Bmal1* expression).

We agree that cells in the lymph nodes, other than T cells, are likely to be contributing to the high amplitude PER2 bioluminescence rhythms in our PMT studies, and had alluded to this in our discussion. We did not intend to use this data to support our hypothesis that Tregs do not have an intrinsic clock. We have now added some text to clarify this point:

“Deletion of *Bmal1* in T cells, which abolishes the clock, did not affect the lymph node clock, suggesting that T lymphocytes do not contribute to the high amplitude rhythms in PER2 bioluminescence we observed from this tissue. Multiple other types of immune cells are found within the lymph nodes, some of which are known to be intrinsically rhythmic (1-3). Furthermore, these cells are held within a capsule of connective tissue, which itself may exhibit rhythmicity. At a cellular level, weak oscillations in clock dependent transcripts have been observed in both naive human (4) and mouse (5) CD4⁺ T cells”

1.3 About the experiments looking at clock gene expression by PCR on Treg RNA (Fig 3b, c), have the authors considered that the sorting procedure might have impacted on clock gene expression? And in

the case of Fig 3b, I would be more cautious with interpretation of the effects of the KO, as only 2 time points were used.

We agree that this is an important consideration. However, other data presented in our paper indicates that the molecular clock still shows robust oscillations in other cell populations which have been sorted, including resident and recruited macrophages (Figure 1a,b) and neutrophils (Figure 1f). For example, expression of *Per2* was 8-fold higher in joint derived MHCII^{low} macrophages sorted at ZT18 versus ZT6. Thus, we are confident that the sorting procedure does not greatly alter clock gene expression. However, to examine the effects of the cell processing associated with the sorting procedure we utilised DEREK mice which express GFP in FoxP3⁺ cells to sort Tregs from lymph nodes at ZT6 and ZT18, thus avoiding staining with fluorescently labelled antibodies and the associated washes and centrifugation steps (Supplementary Figure 5c). In keeping with the rest of our paper, GFP⁺ Tregs did not show any time of day variation in *Per2*, *Bmal1* or *Cry1*, but did show significant diurnal variation in *Rev-erb α* expression (Supplementary Figure 5d). These additional studies confirm that cell staining does not influence clock gene expression.

Regarding the use of two opposing time points in the CD4-Bmal1^{-/-} mice, we concur that this data must be interpreted carefully, and this is why in subsequent experiments with sorted Tregs we utilised 4 time points over 24 hours (Figure 3c) to further confirm lack of rhythmicity in *Per2*, *Bmal1* and *Cry 1* expression in Tregs.

2. Based on the previous point (absence of Treg-intrinsic clock regulation), the authors address possible external timing cues. Based on previous circadian T cell literature (which the authors cite), glucocorticoids and CXCR4 regulation were obvious candidates to test. The authors do this by looking at CXCR4 expression on Tregs, and then by exogenous administration of dexamethasone. These are nice experiments and the results are interesting. However, although they are consistent with the model, these data do not prove that this is the only or even the main mechanism for circadian regulation of Treg in their model, as stated in different places in the manuscript (e.g., line 269-270, line 314).

The reviewer raises a good point, and we have been very cautious in the interpretation of our data and we state that "Despite the rhythmic behaviour of Tregs, they lack a functional, cell-autonomous circadian clock, and therefore respond to systemic timing cues, *including* the hypothalamic-pituitary-adrenal (HPA) axis". Furthermore, we say that "We now show that dexamethasone drives CXCR4 expression in Tregs, implicating activation of the glucocorticoid receptor as *sufficient* to impart a circadian signal". Hence, although we demonstrate circadian responsiveness of Treg trafficking we do not claim that this is the only (or main) mechanism driving diurnal regulation of Treg function. We agree that there are several other potential rhythmic signals that could contribute to the rhythmic phenotype we have observed, but it is beyond the limits of this manuscript to explore them here.

3 One very interesting aspect of the model put forward by the authors is that the Tregs confer rhythmicity in the CIA model via an effect on myeloid cells, in particular the Ly6Chi monocytes. Unfortunately, this part of the study lacks firm demonstration.

3.1 The authors write (line 225-226): "These data confirm a role for Tregs in conferring a time of day variation in arthritis severity." However, the report is lacking experiments specifically comparing disease progression or paw inflammation after CIA at ZT6 vs 18, in the context of Treg depletion. Such data would strengthen the manuscript.

It is well established that Tregs play a protective role in the pathogenesis of arthritis. Adoptive transfer of antigen-specific Tregs impedes progression of established arthritis (6). Furthermore, it has already been demonstrated using antibody depletion approaches and DEREK mice that depletion of Tregs in murine models of arthritis perpetuates disease progression (7) (8) (9). Our aim here was not to address how disease progression is affected by Treg depletion (which is already established) but to ask how Tregs contribute to diurnal variation in inflammation within joints. We designed our experiments with the DEREK mice to be acute studies, to address specifically the effects of full Treg depletion on the diurnal variation in cytokine expression within inflamed joints. We have now

amended our text to reflect what has already been shown in the field regarding the impact of Treg depletion on disease progression:

"It is well established that Tregs play a protective role in the pathogenesis of arthritis. Adoptive transfer of antigen specific Tregs impedes progression of CIA (6) whilst depletion of Tregs perpetuates disease progression, increasing disease score and paw swelling (7-9). To test the hypothesis that the increase in abundance of these repressive cells at night confers time of day variation in disease markers we utilised DERE mice, in which Tregs are depleted by administration of diphtheria toxin (DTX)."

3.2 Also, the link between the Treg rhythms and the monocytes is based only on the impact of Treg depletion on monocyte counts and IL1beta expression, but the effect could be indirect.

The reviewer raises an interesting point. We clearly outlined altered expression of IL1 β by Ly6C^{hi} monocytes in the absence of Tregs. Moreover, there are previous reports of Treg suppression of monocyte function *in vitro* (10-12) and a more recent publication which observed a similar impact of Tregs on Ly6C^{hi} monocytes in the skin in the setting of psoriasis (we have amended our manuscript to include a reference to this paper (13)). However, the reviewer is correct in that although this interaction between Tregs and Ly6C^{hi} monocytes is well documented the mechanism(s) by which Tregs impact monocyte function remain unclear and could well be indirect. We have added the below discursive statements to our manuscript to address this intriguing point

"Our studies indicate that within the inflammatory milieu of the joint, Tregs repress expression of IL1 β by Ly6C^{hi} monocytes. This observation is supported by several *in vitro* studies (10-12) and recent studies in the setting of psoriasis where Tregs restrain the pro-inflammatory action of Ly6C^{hi} cell populations in the skin(13). The mechanistic nature of this interaction (cytokine mediated or contact dependent) is as yet unclear."

3.3 Moreover, the authors wrote in the Discussion (line 342-344): "It is highly likely that the cellular targets for the repressive action of Tregs extend beyond these myeloid cells, to include T effector cells, but we were unable to explore this with our assays". Why do they think that the picture is incomplete? Why could this question not be explored? These doubts expressed by the authors support my concerns raised above, that the Treg-myeloid connection might be either indirect or only a part of the mechanism.

The reviewer is correct that the impact of Tregs on Ly6C^{hi} monocytes could only be part of the mechanisms contributing to circadian controlled Treg-dependent changes in inflammation severity. However, in order to be able to explore circadian variation in the suppressive action of Tregs on T cells we would need to perform *ex vivo* assays, which is technically impossible for the following key reasons. Treg suppression assays typically require co-culture of Tregs with CD4⁺ or CD8⁺ cells for 72h(14). Therefore when considering the difference in activities over 24h, this time frame is not appropriate.

3.4 In addition to this, how the synoviocytes (and their clock) affect or mediate these processes (including the involvement of Tregs) is unclear and should be addressed.

The reviewer raises interesting comments regarding the role of fibroblast-like synoviocytes (FLS) in regulating rhythmic Treg function. It is now established that FLS exist as multiple anatomically distinct subsets, each contributing differently to the inflammation and tissue damage associated with arthritis (15-17). As indicated by the reviewer, these resident joint mesenchymal cells are established to have an intrinsic clock (18-21). Furthermore, disruption of this clock, via *Bmal1* deletion causes enhanced pro-inflammatory responses to local inflammation (18). It is conceivable that these joint resident cells could provide a rhythmic signal which may drive recruitment or retention of immune cells (such as Tregs) to the inflamed joints.

In terms of recruitment of Tregs to inflamed joints, we must consider tissue specific mechanisms driving Treg trafficking to the site of inflammation. It is not clear which chemotactic signals drive Treg trafficking to inflamed joints. Whilst Tregs in the periphery have been well characterised in RA patients (22), few studies have phenotyped Tregs within the synovium. Jiao and colleagues report enhanced expression of a subset of chemokine receptors (CCR4, CCR5 and CXCR4) on Tregs isolated from the synovial fluid of patients with active RA compared to Tregs isolated from peripheral blood. (23) In the CIA model, it has been shown that CCR2 blockade during established arthritis aggravates clinical signs of arthritis, which is suggested to be through interference with the ability of Tregs to migrate to the site of inflammation (24). Ligands for these receptors include CCL2 (CCR2); CCL3, CCL4 and CCL5 (CCR5); CCL17 and CCL22 (CCR4) and CXCL12 (CXCR4). Ourselves and others have shown that FLS can produce several of these chemokines. For example, FLS produce abundant levels of CCL2 and CCL5 (but not CCL3 or CCL4) in response to TNF α stimulation, and constitutively express CXCL12 (25) (**Figures for Reviewers 1**). Thus, FLS may be a viable source of a chemotactic signal driving rhythmic recruitment of Tregs to the synovium. However, it may be that levels of the chemokine itself do not need to be rhythmic if expression of the receptor is (as in the instance of CXCR4-CXCL12). Without a clearer understanding of the mechanisms driving Treg recruitment to the joints, we cannot viably investigate the role of the FLS in this.

There is some evidence to suggest that FLS may affect the retention of Tregs within the joints. *In vitro* co-culture studies demonstrate that FLS derived from arthritic animals downregulate FoxP3 expression on Tregs via a contact dependent mechanism involving the interaction of GITR ligand (TNFSF18) on FLS and the receptor (GITR) on Tregs (26). Whilst expression of GITR on Tregs was up-regulated in arthritic mice, expression levels did not vary by time of day in either naïve or arthritic mice (**Figure 2e**). We have now assessed GITRL expression by FLS. Analysis of gene expression by Q-PCR reveals no change in the transcript of *Gitrl* in response to stimulation via TNF α or LPS (**Figures for Reviewer 2**). Furthermore, after 24h stimulation we could not detect production of GITRL protein (ELISA kit DY2177 Mouse GITR Ligand (TNFSF18) R&D Systems). Subsequently, in our hands we have no evidence that FLS regulate Treg retention in joints via GITR-GITRL interactions.

We now allude to these potential interactions and mechanisms in the manuscript:

“These data suggest that rhythmic extrinsic factors regulate Treg activity within the joint. These rhythmic signals may be locally derived in the joint or emanate from elsewhere, and could facilitate Treg recruitment to or retention within the joints. Fibroblast-like synoviocytes (FLS) play a key role in the pathogenesis of inflammatory arthritis (27), coordinating local inflammation through the secretion of cytokines and chemokines. It is now established that FLS exist as multiple anatomically distinct subsets, each contributing differently to the inflammation and tissue damage associated with arthritis (15-17). FLS are circadian rhythmic (18-20) and present as a candidate source for a locally derived rhythmic signal. Mechanisms driving Treg recruitment to arthritic joints are not well described. Tregs isolated from the synovial fluid of RA patients express higher levels of CCR4, CCR5 and CXCR4(27), which together have a multitude of chemokine ligands (including CCL17, CCL22, CCL16, CCL3, CCL3L1, CCL4, CCL5, CCL14 and CXCL12 (28)), some of which are established to be produced by FLS (15). The role of these rhythmic resident synovial fibroblast populations in directing rhythmic Treg biology warrants further investigation, but is beyond the scope of this study.”

We agree with the reviewer that how clocks within joint resident FLS may regulate the rhythmic behaviour of Tregs is worthy of investigation. Whilst we have generated mice lacking a functional clock in FLS (*Col6a1-Bmal1*^{-/-}) (18) our testing indicates that this transgenic mouse line is unfortunately not susceptible to collagen induced arthritis (CIA) induced with either chicken or bovine collagen. So, currently this is not something we can test.

4 It would be interesting that the Discussion address the results in the context of other literature on rodent experimental models of arthritis. The authors cite their previous studies (Hand et al, 2016, 2019), but it is surprising that there is no discussion at all in the context of the Hand et al 2019 paper, which showed a role of the clock in mesenchymal cells to control the disease. This is particularly interesting in the context of the current manuscript, which suggests that the Treg circadian regulation

is not relying on a clock within Tregs. Studies by Hashiramoto et al, JI, 2010, and Yoshida et al, Scand J Rheumatol, 2013, should also be discussed.

We thank the reviewer for this suggestion and have now added to the discussion reference to previous studies using rodent models of arthritis to demonstrate a role for the clock in disease pathogenesis:

“Circadian variation in disease severity is frequent, but the mechanisms explaining this are undefined. Circadian disruption through genetic targeting of the core clock genes *Cry1/2* or *Bmal1* is associated with aggravated disease in the more simplistic mouse model of arthritis, collagen antibody induced arthritis(18, 29), providing evidence for the role of the clock in restraining the pathogenesis of this disease.”

We also now refer to the work of Yoshida *et al* in a section referring to the effects of chronic inflammation on intrinsic clocks within synovial cells.

“Analysis of joint-derived macrophages and neutrophils revealed significant dampening of intrinsic clocks in these inflammatory cells under arthritic conditions. In keeping with *in vitro* studies, *Rev-erb* α expression was highly sensitive to inflammation (30). Earlier work has identified changes in the expression of components of the core clock within synovial fibroblasts in the setting of chronic arthritis (19, 20, 31, 32), but this is to our knowledge the first observation of the effects of chronic joint inflammation on the intrinsic clock within macrophages and neutrophils. Whether the amplitude of residual intrinsic oscillations is sufficient to maintain circadian control of immune activities of these cells remains to be determined.”

More specific concerns:

5. Conclusions about a dampening of the macrophage and neutrophil clocks in the inflamed joints are a bit overstated given that the data are only for two time points. Moreover, in the case of the neutrophils, although *Reverba* expression is reduced, the other clock genes seem to retain a time-dependent variation.

We have been cautious in our interpretation of this data and state in the discussion: “analysis of joint derived macrophages and neutrophils revealed significant dampening of intrinsic clocks ... under inflammatory conditions. ... Whether the amplitude of residual intrinsic oscillations is sufficient to maintain circadian control of immune cells remains to be determined”. We feel this is an accurate interpretation of the data.

6. It is good that the FACS gating strategy is shown for some stainings, but this should be the case for all stainings. Examples of FACS plots and gating strategy should be shown for all stainings, including CD4/CD8 T cells (suppl. fig. 2), Treg markers (Fig 2d).

As requested, we have now included -

Supplementary Figure 2a: The gating strategy for T cell subsets (CD4⁺, CD8⁺, Th1, Th17 and Tregs)

Supplementary Figure 2b: The gating strategy for Treg activation/proliferation markers

Additionally, all new data is accompanied by the relevant gating strategy.

7. About NRP1 expression (Fig 2d), it is said that it is higher in the night, but the figure seems to show a non significant difference, although the lack of significance could be due to a lack of statistical power, and a increase in the group size might make it significant.

We have now corrected this figure, as the statistics were omitted in our previous version. The 2-way ANOVA of NRP1 expression in Tregs shows a significant interaction between time-of-day and arthritis (P=0.0285). [But no significant effect of time-of-day (P=0.3093) or arthritis (P=0.1045).] Post-hoc analysis (Bonferonni’s multiple comparison test) identifies significant differences in NRP1 expression between ZT6 and ZT18 within arthritic mice (P=0.0464) (but not controls, P>0.0999). Furthermore,

post-hoc analysis identifies a significant increase in NRP1 expression in arthritic mice compared to controls at ZT18 (P=0.0371), but not at ZT6 (P=0.3145).

8. Please check the whole manuscript for the format of gene/transcript names, and remain consistent (capital letter or not; mouse genes should have first letter cap).

We thank the reviewer for pointing out our inconsistency here and have corrected all gene names as requested.

9. Fig 4b has a problem with the indication of stats: the difference is between time points, not genotypes.

We have now repeated this experiment using the same biotin labelled CXCR4 antibody as utilised in Figure 4d. To assess time-of-day variation in CXCR4 expression and any effects of deletion of the clock in T cells a 2-way ANOVA and post hoc Tukey test was utilised. Both inguinal lymph nodes and spleen showed a significant time-of-day effect (but no genotype difference).

10. Regarding Fig. 4d, it is said (line 194) that arthritic mice are "similar to controls". This is vague. What do the stats say? Is there an interaction in the ANOVA? An effect of genotype?

We apologise to the reviewer for our lack of clarity and have now added more detail on the output of the statistical comparisons performed as follows:

“Serial blood sampling from naïve and arthritic mice revealed that diurnal variation in serum corticosterone levels persist in the setting of chronic inflammation, with peak concentration between ZT6 and ZT12 (Figure 4e). Statistical analysis revealed a significant interaction between time-of-day and treatment, and a significant effect of time-of-day on corticosterone levels. Post hoc analysis revealed no significant differences between treatment groups at any time point, but significant increases above ZT0 values at ZT6 (arthritic only) and ZT12 (control and arthritic). This suggests the influence of rhythmic glucocorticoids on Treg function could persist even under chronic inflammatory conditions.”

11. Why are some graphs with clouds of dots (one dot per sample) while other graphs are bar graphs? Clouds of dots give a more direct indication of the actual data and the variability, and should be preferred, I think.

We have now amended all the graphs in the results to show each individual data point as requested.

12. About suppl. fig 5c, it is said that pro-inflammatory cytokines are more expressed at ZT6 than at ZT18. The stats do not support this for 4 out of 6 cytokines/chemokines. The group size should be increased to have more statistical power?

We have now amended the text to reflect this as follows:

“Furthermore, in keeping with our observations in DBA/1 mice (33), expression of pro-inflammatory cytokines in affected limbs was heightened at ZT6 versus ZT18, but this only reached statistical significance for *Cxcl1* and *il6* (Supplementary Figure 7c).”

The group sizes here were 7/group. We agree that this trend may become significant if greater numbers were used, but based on the variance very large numbers of mice would be needed, and the effect might lack biological significance.

13. Line 213 says that the disease progresses both in control and Treg-depleted mice, but Fig 5b does not show a statistically significant progression.

In the earlier version of Figure 5b a paired T-test of the data from the “depleted” group revealed that although paw thickness increased in each animal over this period of time (ranging from 6-22%

increase in thickness) this change did not reach a statistically significantly different ($P=0.0569$). In order to support our statement that disease progressed in both control and Treg depleted mice, we have now added more data to this figure from a second cohort of animals. We now plot the paired data for each animal and show that in both our control and depleted groups the thickness on Day 3 was significantly greater than the thickness on Day 1. We would like to also note that the paw thickness only reflects disease progression in the hind paws, and does not reflect swelling of the fore paws (which cannot be measured using this technique).

14. In Supplementary Fig 6a, are the F4/80⁻ cells really negative? How is the gate set?

We routinely utilise full minus one (FMO) control staining when we set up any new panel of antibodies for flow cytometry. This ensures that we are only selecting truly positively stained cells to quantify and/or take forward to the next gate. As an example (and to answer the reviewer's direct query) please see **Figure for Reviewer 3** which shows data from the set-up of the staining panel used in Supplementary Figure 8a. This is validation of our gating strategy for identifying F4/80⁺ macrophages, and therefore the F4/80⁻ cells.

15. Fig 5g: A 2-way ANOVA should be performed, not 1-way.

We thank the reviewer for spotting our error and have now performed a 2-Way ANOVA and post-hoc Bonferonni test on this data, and report these results in the figure. This has not changed the interpretation of our results.

16. It is good that the authors have provided a table of the antibodies used in the study, but the fluorophores and suppliers should be added.

We have now amended the table to include these extra details (**Supplementary Table 1**)

17. For the QPCRs, the TaqMan probe sets which were used should be listed. How the two control genes were used should also be described (average of Cts?).

We have now included in the supplementary figures a table of primers and probes used for QPCR (**Supplementary Table 2**). We now indicate in each figure legend which housekeeping gene was utilised.

Reviewer #2 (Remarks to the Author):

The authors make the novel observation that Tregs, as a critical immune cell population, observe time-of-day differences in inflamed arthritic joints. The authors then go on to show that absence of Tregs ablates time-of-day rhythmicity in inflammatory parameters. While the findings is very interesting and generally well demonstrated, the explanation of altered Treg numbers and function in joints should be developed more.

Main points:

1. Suppl. Fig. 2 CD4 and Fig. 2b and C: There is in the CIA condition a big increase in the CD4 compartment at ZT18 (3 fold higher compared to ZT6) yet, Th1 and Th17 are not affected. The only increase is seen in Tregs. Are the authors saying that these are the predominant populations of CD4 T cells in this scenario? It would be important then to show absolute numbers of the total and individual CD4 populations to support that claim.

The reviewer is correct, within inflamed limbs, Tregs make up the predominant population of CD4⁺ cells comprising approximately 40-60% of all CD4⁺ T cells. We have now illustrated this fact in the **Supplementary Figure 2a**. The data presented on Figures 2a-c are combined from 3 independent trials, and consequently cell numbers are presented as % change in order to normalise the data between experimental cohorts. (We occasionally experience variation in total cell counts between experimental cohorts due to batch effects of the collagenase enzyme, thus it was not possible to

present the data as total cell counts). When we look at these experimental runs independently (**Figure for reviewer 4**) it is clear that the increase in CD4⁺ cells in arthritic limbs at ZT18 is a consequence of an increase in total numbers of Tregs. We have now added some additional information to the text to make clear the abundance of Tregs within the CD4+ population.

“Analysis using a further panel of T cell markers revealed that it was CD4⁺ (but not CD8⁺) T cells that showed diurnal variation (Supplementary Figure 2). Of the CD4⁺ populations analysed, Tregs were the most abundant, making up 40-60% of CD4⁺ cells within inflamed joints (**Supplementary Figure 2a**).

2. Fig. 2d: With respect to the proliferation of Ki67+ cells, since the time-of-day-difference is stronger in untreated mice and the level of Ki67+ cells is comparable between untreated and arthritic mice at ZT18, why are Treg numbers not increased at ZT18 in the steady state? Since the etiology of higher Treg numbers is important for the paper the authors should also consider performing additional experiments with BrdU/Edu to verify that Tregs have indeed proliferated. Are the cells recruited from blood or do they proliferate locally? Page 6, line 124-125: ‘... as expression at ZT6 was enhanced by inflammation.’ Is this the case, i.e. is expression of Ki67 enhanced on a per cell basis or are the more cells positive for Ki67?

The reviewer raises some interesting points regarding the etiology of the additional Tregs within the joints at night. As suggested, we have now performed further *in vivo* studies with EdU labelling of joint derived cells, which detects DNA synthesis in proliferating cells during the S phase of the cell cycle (**Figure 2c**). These data indicate a trend for increased EdU positive Tregs in arthritic mice compared to naïve mice, which only reaches statistical significance at ZT6. We note that although the Ki67+ and/or EdU+ Tregs we detect have recently proliferated, this is not evidence that they have proliferated within the joint. To fully address this, we would have to carry out more refined tracking and tracing studies. As the reviewer points out increases in numbers of Ki67+ (and EdU+) Tregs do not correlate to increased number of total Tregs within the joints. Consequently, this data supports a narrative that the increased abundance of Tregs within the joints at ZT18 is a consequence of trafficking or infiltration, rather than local proliferation. We have now modified the text of the manuscript to include the EdU data and to reflect these thoughts:

“Further analysis examined the activity and proliferation of Tregs within the inflamed joints at different phases of the day (**Figure 2d,e and Supplementary Figure 3**). Analysis of Ki67 expression showed a time of day effect in naïve mice, with more recently proliferated cells at ZT18. Conversely EdU staining (which marks cells undergoing S-phase) did not show the same effect. EdU labelling studies did reveal increased incorporation of EdU into Tregs from arthritic animals compared to naïve animals at ZT6. However, from both Ki67 and EdU staining we did not observe an increase in Tregs from inflamed mice at ZT18, suggesting that the increased Treg numbers at ZT18 was not a consequence of elevated Treg proliferation.”

“Tregs isolated from arthritic joints at ZT18 did not show enhanced signs of recent proliferation, suggesting that increased numbers within the joints at night are more likely a consequence of recruitment. In keeping, expression of the chemokine receptor CXCR4 was increased on naïve Tregs during the dark-phase. These data suggest that Tregs are more pro-migratory during the dark-phase (a phenomenon that has been observed in naïve CD4⁺ T cells⁸), and more proliferative, and this may account for increased numbers.”

3. Additionally, it would be important to show that activity of Tregs is heightened at specific times, e.g. using a functional assay, not just a surface staining and/or they should investigate the levels of IL-10 in the joints. NRP1 staining is not significant, at least this is not indicated. Are the stats mislabeled between NRP1 and GITR? These molecules should be explained.

We thank the reviewer for raising the important question of alterations in Treg activity across the 24h period. We explore further the role for IL10, a key Treg suppressive cytokine. We have previously analysed *IL10* transcripts in arthritic joints over the course of 24h, and although these are raised in CIA samples, there were no changes in overall expression over the course of 24h (33). However, IL10 is

likely to be derived from multiple cellular sources within the joints, including T cells, monocytes and macrophages. Therefore, to assess changes in Treg production of IL10 across the 24h period we stimulated Tregs from inflamed joints with PMA and ionomycin. Results confirmed higher numbers of Tregs within arthritic joints at ZT18 versus ZT6, but importantly demonstrated that the capacity of Treg cells to produce IL10 was stable across the two time points. Indicating activity of the Treg *per se* is not altered, more their total numbers. We have now included this data in the manuscript:

“In separate experiments, we examined the production of IL10 by Tregs from arthritic joints at ZT6 and ZT18 by *ex vivo* restimulation (Figure 2f and Supplementary Figure 3). As before, numbers of Tregs in the joints were heightened at ZT18, but the percentage of IL10⁺ Tregs was consistent at both time points.

“However, Tregs harvested from joints at the peak and nadir of disease exhibited the same capacity to secrete IL10 upon stimulation, suggesting and it is the increase in numbers within the joints (rather than an increase in the suppressive activity of each cell) driving daily variation in local inflammation.”

Regarding NRP1 staining – please see correction above (point 7). We have now defined NRP1 and GITR in the main text.

4. As the major inflammatory cell types in the joint (presumably neutrophils and monocytes/macrophages, the absolute numbers of CD45 subpopulations should be shown to be able to discuss this) show no time-of-day difference, what do the authors propose is the cell type that mediates the rhythmic changes with respect to inflammatory swelling and paw thickness? Are the authors saying that it is solely regulated at the anti-inflammatory level by Tregs? DERE (+DTX) mice still show oscillations in Cxcl1 expression, indicating pro-inflammatory cells to be involved as well. This should be discussed more.

The only diurnal change in the cellular composition of the inflammatory environment we detected involved Tregs. Our studies in DERE mice (QPCR analysis and flow cytometry) demonstrate that when present within the joints, these cells actively dampen expression of inflammatory mediators within the local environment. Thus we conclude that alterations in Treg numbers contribute to the rhythmic inflammatory signature observed in arthritis. We cannot conclude that this is the sole mechanism driving rhythmic inflammation. Indeed, it is possible that pro-inflammatory cells within the joints provide a rhythmic inflammatory output (despite evidence that their intrinsic clockwork machinery is disrupted). We have encompassed these thoughts in the discussion as follows:

“Whether the amplitude of residual intrinsic oscillations in pro-inflammatory cells is sufficient to maintain circadian control of immune activities remains to be determined. However, we present a mechanisms by which non-cell autonomous signals contribute to rhythmic repression of inflammation”.

5. Since this is the first time that Tregs have been investigated for circadian clock gene expression, and since this is important for the conclusion of the story, the authors should include possibly a few more clock genes. E.g., from the studies cited, it seems that Per3 was the gene, together with Revba that oscillated the most.

We have also investigated in these samples *Bmal2* (which was undetectable, data not shown) and *Per3* (which was not rhythmic) and have added the *Per3* data to Supplementary Figure 5b.

6. Fig. 5e-f: It would be important to show the data from control and DERE-DTX mice in the same graph to assess whether oscillations are ablated. These seem indeed to be ablated, with the exception of Cxcl1.

The data generated using DERE mice to examine the impact of Treg depletion on inflammatory mediators at different times of day was generated via two separate cohorts of animals. The first where animals were culled at ZT18 and the second where animals were culled at ZT6. Samples from

these two experimental cohorts were analysed by QPCR separately. As such, we apologise but do not think it would be appropriate to pool this data together onto the same graph.

7. What is the (rhythmic?) phenotype of the Cd4Bmal1 KO mice/Tregs in the joints with respect to arthritis? Since the authors have previously published with the Reverba KO mouse and since Reverba is the only rhythmic clock gene: Do these KO mice have rhythms in arthritis?

The reviewer raises some excellent points here. We agree that it would be interesting to examine whether mice lacking *Bmal1* in T cells (CD4-*Bmal1*^{-/-}) still show rhythms in disease activity. However, unfortunately our colony of CD4-*Bmal1*^{-/-} mice (which are on the C57BL/6 background) are not susceptible to CIA in our hands – so this is not something we can currently test. Furthermore, we concur that given our interesting observation that *Rev-erb*α exhibits daily rhythms in naïve Tregs, it would be of interest to examine the contribution of this nuclear receptor to the rhythmic inflammatory action of Tregs. However, REV-ERB plays a critical role in regulating inflammatory pathways within multiple different cell types, including macrophages (36-38), Th17 cells (39) and lung Club cells (30). Therefore, we believe that using the *Rev-erb* α^{-/-} mice in our CIA model would not give us a clear indication of the role of REV-ERB in directing rhythmic function of Tregs.

Minor:

8. Fig 1C: statistics should be shown between control and inflamed groups as all clock genes, including *Bmal1*, are dramatically reduced, which should be noted.

We have now added annotations to **Figure 1C** (macrophages) and **1F** (neutrophils) as suggested showing statistical differences between control and CIA groups at ZT6 and ZT18.

9. What is the gating difference between Suppl Fig. 1 and 6 with respect to inflammatory monocytes? Are the authors missing inflammatory monocytes in Suppl Fig. 1?

Supplementary Figure 1 provides the gating strategy to generate data shown in **Figure 1**. In these earlier experiments we did not pursue monocytes populations in our F4/80⁺ population. **Supplementary Figure 8** refers to the gating strategy to generate data shown in **Figure 5g** (and **Supplementary Figure 8b and c**) where we did investigate monocytes.

10. Fig. Suppl. 2: The fold change normalized to at ZT6 should be 1 not 10.

We thank the reviewer for this observation, and we have corrected the presentation of the data accordingly.

11. What tissues do 'limbs' include, muscle of the leg? Or is it only the joints?

Limbs were dissected from mice 1mm above the ankle or wrist joint and the skin was removed before the tissue was snap frozen. Therefore, this tissue contained bone, joint and muscle. We have added further detail to the Methods section to describe how limbs were harvested from mice.

12. Fig. 2e: The y axis should be labeled.

We apologise to the reviewer for this omission and this has been corrected.

References

1. Silver AC, Arjona A, Walker WE, & Fikrig E (2012) The circadian clock controls toll-like receptor 9-mediated innate and adaptive immunity. *Immunity* 36(2):251-261.
2. Hopwood TW, et al. (2018) The circadian regulator BMAL1 programmes responses to parasitic worm infection via a dendritic cell clock. *Sci Rep* 8(1):3782.
3. Keller M, et al. (2009) A circadian clock in macrophages controls inflammatory immune responses. *Proc.Natl.Acad.Sci.U.S.A* 106(50):21407-21412.
4. Bollinger T, et al. (2011) Circadian clocks in mouse and human CD4+ T cells. *PLoS One* 6(12):e29801.
5. Druzd D, et al. (2017) Lymphocyte Circadian Clocks Control Lymph Node Trafficking and Adaptive Immune Responses. *Immunity* 46(1):120-132.
6. Sun G, et al. (2018) Adoptive Induced Antigen-Specific Treg Cells Reverse Inflammation in Collagen-Induced Arthritis Mouse Model. *Inflammation* 41(2):485-495.
7. Morgan ME, et al. (2003) CD25+ cell depletion hastens the onset of severe disease in collagen-induced arthritis. *Arthritis Rheum* 48(5):1452-1460.
8. Atkinson SM, et al. (2016) Depletion of regulatory T cells leads to an exacerbation of delayed-type hypersensitivity arthritis in C57BL/6 mice that can be counteracted by IL-17 blockade. *Dis Model Mech* 9(4):427-440.
9. Irmiler IM, Gajda M, & Kamradt T (2014) Amelioration of experimental arthritis by stroke-induced immunosuppression is independent of Treg cell function. *Ann Rheum Dis* 73(12):2183-2191.
10. Taams LS, et al. (2005) Modulation of monocyte/macrophage function by human CD4+CD25+ regulatory T cells. *Hum Immunol* 66(3):222-230.
11. Tiemessen MM, et al. (2007) CD4+CD25+Foxp3+ regulatory T cells induce alternative activation of human monocytes/macrophages. *Proc Natl Acad Sci U S A* 104(49):19446-19451.
12. Romano M, et al. (2018) Expanded Regulatory T Cells Induce Alternatively Activated Monocytes With a Reduced Capacity to Expand T Helper-17 Cells. *Front Immunol* 9:1625.
13. Stockenhuber K, et al. (2018) Foxp3(+) T reg cells control psoriasiform inflammation by restraining an IFN- γ -driven CD8(+) T cell response. *J Exp Med* 215(8):1987-1998.
14. Collison LW & Vignali DA (2011) In vitro Treg suppression assays. *Methods Mol Biol* 707:21-37.
15. Croft AP, et al. (2019) Distinct fibroblast subsets drive inflammation and damage in arthritis. *Nature* 570(7760):246-251.
16. Mizoguchi F, et al. (2018) Functionally distinct disease-associated fibroblast subsets in rheumatoid arthritis. *Nat Commun* 9(1):789.
17. Zhang F, et al. (2019) Defining inflammatory cell states in rheumatoid arthritis joint synovial tissues by integrating single-cell transcriptomics and mass cytometry. *Nat Immunol* 20(7):928-942.
18. Hand LE, Dickson SH, Freemont AJ, Ray DW, & Gibbs JE (2019) The circadian regulator Bmal1 in joint mesenchymal cells regulates both joint development and inflammatory arthritis. *Arthritis Res Ther* 21(1):5.
19. Haas S & Straub RH (2012) Disruption of rhythms of molecular clocks in primary synovial fibroblasts of patients with osteoarthritis and rheumatoid arthritis, role of IL-1 β /TNF. *Arthritis Res Ther* 14(3):R122.

20. Kouri VP, *et al.* (2013) Circadian timekeeping is disturbed in rheumatoid arthritis at molecular level. *PLoS One* 8(1):e54049.
21. Becker T, *et al.* (2014) Clock gene expression in different synovial cells of patients with rheumatoid arthritis and osteoarthritis. *Acta Histochem* 116(7):1199-1207.
22. Walter GJ, *et al.* (2016) Phenotypic, Functional, and Gene Expression Profiling of Peripheral CD45RA+ and CD45RO+ CD4+CD25+CD127(low) Treg Cells in Patients With Chronic Rheumatoid Arthritis. *Arthritis Rheumatol* 68(1):103-116.
23. Cheng WC, *et al.* (2016) Periodontitis-associated pathogens *P. gingivalis* and *A. actinomycetemcomitans* activate human CD14(+) monocytes leading to enhanced Th17/IL-17 responses. *Eur J Immunol* 46(9):2211-2221.
24. Bruhl H, *et al.* (2004) Dual role of CCR2 during initiation and progression of collagen-induced arthritis: evidence for regulatory activity of CCR2+ T cells. *J Immunol* 172(2):890-898.
25. Bradfield PF, *et al.* (2003) Rheumatoid fibroblast-like synoviocytes overexpress the chemokine stromal cell-derived factor 1 (CXCL12), which supports distinct patterns and rates of CD4+ and CD8+ T cell migration within synovial tissue. *Arthritis Rheum* 48(9):2472-2482.
26. Kim SH & Youn J (2012) Rheumatoid Fibroblast-like Synoviocytes Downregulate Foxp3 Expression by Regulatory T Cells Via GITRL/GITR Interaction. *Immune Netw* 12(5):217-221.
27. Bartok B & Firestein GS (2010) Fibroblast-like synoviocytes: key effector cells in rheumatoid arthritis. *Immunol Rev* 233(1):233-255.
28. Yoshie O & Matsushima K (2015) CCR4 and its ligands: from bench to bedside. *Int Immunol* 27(1):11-20.
29. Hashiramoto A, *et al.* (2010) Mammalian clock gene Cryptochrome regulates arthritis via proinflammatory cytokine TNF-alpha. *J Immunol*. 184(3):1560-1565.
30. Pariollaud M, *et al.* (2018) Circadian clock component REV-ERBalpha controls homeostatic regulation of pulmonary inflammation. *J Clin Invest* 128(6):2281-2296.
31. Yoshida K, *et al.* (2013) TNF-alpha modulates expression of the circadian clock gene *Per2* in rheumatoid synovial cells. *Scand J Rheumatol* 42(4):276-280.
32. Yoshida K, *et al.* (2018) TNF-alpha induces expression of the circadian clock gene *Bmal1* via dual calcium-dependent pathways in rheumatoid synovial cells. *Biochem Biophys Res Commun* 495(2):1675-1680.
33. Hand LE, *et al.* (2016) The circadian clock regulates inflammatory arthritis. *FASEB J* 30(11):3759-3770.
34. Bishehsari F, *et al.* (2019) Alcohol Effects on Colon Epithelium are Time-Dependent. *Alcohol Clin Exp Res* 43(9):1898-1908.
35. Fredrich M, Hampel M, Seidel K, Christ E, & Korf HW (2017) Impact of melatonin receptor-signaling on Zeitgeber time-dependent changes in cell proliferation and apoptosis in the adult murine hippocampus. *Hippocampus* 27(5):495-506.
36. Sato S, *et al.* (2014) Direct and indirect suppression of interleukin-6 gene expression in murine macrophages by nuclear orphan receptor REV-ERBalpha. *ScientificWorldJournal* 2014:685854.
37. Gibbs JE, *et al.* (2012) The nuclear receptor REV-ERBalpha mediates circadian regulation of innate immunity through selective regulation of inflammatory cytokines. *Proc Natl Acad Sci U S A* 109(2):582-587.
38. Sitaula S, Billon C, Kamenecka TM, Solt LA, & Burriss TP (2015) Suppression of atherosclerosis by synthetic REV-ERB agonist. *Biochem Biophys Res Commun* 460(3):566-571.
39. Amir M, *et al.* (2018) REV-ERBalpha Regulates TH17 Cell Development and Autoimmunity. *Cell Rep* 25(13):3733-3749 e3738.

Reviewers' comments:

Reviewer #1 (Remarks to the Author):

I thank the authors for their careful revision in response to my comments. The manuscript has been improved by these revisions. I only have a number of remaining specific comments.

1) My main remaining concern relates to the first concern of my previous review: I still think that the authors' conclusions about the lack of clock in Treg cells are too definite. The PER2::LUC LN data are not conclusive about this, the two time point gene expression could hide an effect due to the inappropriate selection of time points, and the four time point experiments, although better, could be unable to detect slight clock gene oscillations. About the latter point, it should be noted that previous report showed low amplitude but self-sustained bioluminescence rhythms in human CD4+ T cells (Bollinger et al, PLOS ONE, 2011) and mouse CD8+ T cells (Nobis et al, PNAS, 2019). Also, the Hemmers & Rudensky 2015 paper showed clock gene rhythms in CD4+ T cells (even if of lower amplitude than in the liver). Although Hand et al's data support a model where Treg count rhythms is regulated in a diurnal fashion by exogenous cues, I would tone down the conclusion that they lack a clock. More specifically:

- Line 33: "Finally, we identify that Tregs are not intrinsically circadian rhythmic": maybe write: "Finally, Tregs do not appear to be intrinsically circadian rhythmic..."

- Line 35: "... we report a novel circadian rhythmic network in which non-rhythmic cells, Tregs, are driven to rhythmic activity...": maybe write: "... our data support a model in which non-rhythmic cells, Tregs, are driven to rhythmic activity..."

- Line 82: "...despite not having cell-autonomous circadian clocks": maybe write: "...despite having no clear overt intrinsic circadian rhythmicity."

2) Fig. 1e, f: I suggest indicating "Neutrophils" at the top of the graphs (as in other panels).

3) Line 107: "Rev-erba rhythms": Given that it is only two time points, "effect of time" would be more appropriate than "rhythm".

4) Line 136: There seems to be a problem with the indication of stats for GITR in Fig 2e: the figure indicates only an effect of time, whereas the text mention that there is no effect of time but rather an effect of arthritis (which is indeed what the data seem to tell).

5) Line 141: "the percentage of IL10+ Tregs was consistent at both time points". The phrasing is misleading or unclear. I guess that the authors meant that the percentage "changed accordingly" (i.e. that it changed in a similar fashion to Tregs).

6) Line 168: "up-regulation of Cry1 as expected". Maybe the authors should include a reference to indicate why this is expected.

7) Line 182: Is reference 25 right here? I cannot see why in this Berod et al paper relates to this sentence. Should it be the Hemmers paper instead (ref. 24)?

8) Lines 190-193: What are the two peaks of bioluminescence? The first one is probably an acute response of Per2 to the stimulation, but could the 2nd peak represent a circadian peak? The authors' interpretation of these data (lines 192-3) is vague.

9) Line 202: "CXCR4 showed rhythms": Given that it is only two time points, "effect of time" would be more appropriate than "rhythm".

10) Line 309: "...they lack a (...) clock, and therefore respond to systemic timing cues...": I don't think that the use of the word "therefore" is appropriate here. Being regulated by a local clock and by systemic cues are not mutually exclusive possibility, such that the systemic regulation is not a

consequence of the lack of a local clock, as the word "therefore" seems to imply. I suggest replacing it by "but on the other hand" or something alike.

11) Line 338: In the previous round of review, I had commented on the interpretation of the LN bioluminescence data. I don't think that the authors' revisions fully address my concern. I don't think that it is right to firmly conclude that T cells do not contribute to LN bioluminescence rhythms. It could be due to an incapacity of the assay to detect the reduction of amplitude in the KOs vs WT LNs. I would add the word "significantly" in this sentence: "... that T lymphocytes do not contribute significantly to the high amplitude rhythms..."

12) Lines 373-384: While the authors cite the papers from Shimba et al (ref 8) and Dimitrov et al (ref 48) in their discussion of the role of glucocorticoids in T cell function and trafficking, they do not mention that both of these papers showed that CXCR4 expression in T cells is regulated by glucocorticoids. This should be described.

13) The authors responded to a previous comment of mine, saying that "...we do not claim that this is the only (or main) mechanism driving diurnal regulation of Treg function. We agree that there are several other potential rhythmic signals that could contribute to the rhythmic phenotype we have observed, but it is beyond the limits of this manuscript to explore them here." I think that this should be mentioned in the Discussion.

14) Lines 610-614: Description of ANOVAs here is fine, but I think that the p for the interaction and/or the main effects should be indicated in the figure legends. This was done by the authors in some cases, but not systematically.

Nicolas Cermakian

Reviewer #2 (Remarks to the Author):

The authors have addressed my concerns.

Response to Reviewers' comments

We would like to thank the two reviewers for taking the time to carefully consider our revised manuscript. We have edited the manuscript in response to comments provided by Reviewer #1. These changes are listed below.

Reviewer #1 (Remarks to the Author):

I thank the authors for their careful revision in response to my comments. The manuscript has been improved by these revisions. I only have a number of remaining specific comments.

1) My main remaining concern relates to the first concern of my previous review: I still think that the authors' conclusions about the lack of clock in Treg cells are too definite. The PER2::LUC LN data are not conclusive about this, the two time point gene expression could hide an effect due to the inappropriate selection of time points, and the four time point experiments, although better, could be unable to detect slight clock gene oscillations. About the latter point, it should be noted that previous reports showed low amplitude but self-sustained bioluminescence rhythms in human CD4+ T cells (Bollinger et al, PLOS ONE, 2011) and mouse CD8+ T cells (Nobis et al, PNAS, 2019). Also, the Hemmers & Rudensky 2015 paper showed clock gene rhythms in CD4+ T cells (even if of lower amplitude than in the liver). Although Hand et al's data support a model where Treg count rhythms is regulated in a diurnal fashion by exogenous cues, I would tone down the conclusion that they lack a clock.

More specifically:

- Line 33: "Finally, we identify that Tregs are not intrinsically circadian rhythmic": maybe write:

"Finally, Tregs do not appear to be intrinsically circadian rhythmic..."

- Line 35: "... we report a novel circadian rhythmic network in which non-rhythmic cells, Tregs, are driven to rhythmic activity...": maybe write: "... our data support a model in which non-rhythmic cells, Tregs, are driven to rhythmic activity..."

- Line 82: "...despite not having cell-autonomous circadian clocks": maybe write: "...despite having no clear overt intrinsic circadian rhythmicity."

We thank the reviewer for his thoughts and suggestions here. We are happy to tone down the conclusions somewhat. Whilst we have no evidence for a functional intrinsic oscillator within Tregs, we appreciate that others have elegantly demonstrated the importance of intrinsic oscillators in other subsets of T cells. We now acknowledge recent work regarding the existence of an intrinsic oscillator with CD8⁺ T cells (Nobis *et al.* PNAS 2019) and have amended our text as suggested above.

2) Fig. 1e, f: I suggest indicating "Neutrophils" at the top of the graphs (as in other panels).

We have added this label to Figure 1e and 1f, which has improved the clarity of the figure.

3) Line 107: "Rev-erba rhythms": Given that it is only two time points, "effect of time" would be more appropriate than "rhythm".

We agree that this is more appropriate and have modified the text to "Analysis of clock genes within neutrophils also revealed loss of the effect of time-of-day on *Rev-erba* expression."

4) Line 136: There seems to be a problem with the indication of stats for G1TR in Fig 2e: the figure indicates only an effect of time, whereas the text mentions that there is no effect of time but rather an effect of arthritis (which is indeed what the data seem to tell).

We apologise for this. The indication on the figure was supposed to represent the fact that there is a significant effect of disease on GTR levels. However, this is obviously not very clear. We have now modified this figure to annotate the graph itself to indicate that there is an effect of disease.

5) Line 141: "the percentage of IL10+ Tregs was consistent at both time points". The phrasing is misleading or unclear. I guess that the authors meant that the percentage "changed accordingly" (i.e. that it changed in a similar fashion to Tregs).

We have now clarified in the text how we have interpreted this data as follows: "Total IL10⁺ Treg numbers in the joints were increased at ZT18 compared to ZT6. By quantifying the percentage of Tregs present within the joints at these times which are IL10⁺ we established that the capacity to secrete IL10 remains consistent between time points. This indicates that it is the increase in overall numbers of Tregs at ZT18, rather than a change in their individual suppressive activity, that drives daily variation in local inflammation."

6) Line 168: "up-regulation of Cry1 as expected". Maybe the authors should include a reference to indicate why this is expected.

We thank the reviewer for this suggestion and have now added reference here to the Kondratov *et al* paper (FASEB J 2006) which demonstrates that CRY represses CLOCK/BMAL transactivation of the *Cry1* promoter.

7) Line 182: Is reference 25 right here? I cannot see why in this Berod *et al* paper relates to this sentence. Should it be the Hemmers paper instead (ref. 24)?

We thank the reviewer for pointing this mistake out. It was an incorrect citation and we have replaced it with Hemmers and Rudensky (2015) which refers to how oscillations in *Rev-erb α* transcription may be maintained in the absence of BMAL1 by extrinsic circadian factors.

8) Lines 190-193: What are the two peaks of bioluminescence? The first one is probably an acute response of Per2 to the stimulation, but could the 2nd peak represent a circadian peak? The authors' interpretation of these data (lines 192-3) is vague.

We have now added our interpretation of this data in the results section as follows: "Stimulation (anti-CD3ε/anti-CD28) caused an induction of PER2 bioluminescence (**Figure 3d**) indicating direct coupling of core clock gene expression to extrinsic T cell stimulation. A second peak was seen approximately 20h later. It is unlikely that this second peak represents circadian activity as Tregs lacking *Bmal1* showed a similar PER2 induction after stimulation (**Supplementary Figure 6b and c**). Instead this may be a consequence of the increase in cell numbers as they undergo proliferation in the expansion media."

9) Line 202: "CXCR4 showed rhythms": Given that it is only two time points, "effect of time" would be more appropriate than "rhythm".

We agree with this comment and have changed the text to "CXCR4 expression showed time-of-day variation".

10) Line 309: "...they lack a (...) clock, and therefore respond to systemic timing cues...": I don't think that the use of the word "therefore" is appropriate here. Being regulated by a local clock and by systemic cues are not mutually exclusive possibility, such that the systemic regulation is not a

consequence of the lack of a local clock, as the word "therefore" seems to imply. I suggest replacing it by "but on the other hand" or something alike.

We agree that the two control mechanisms are not mutually exclusive. We have altered the sentence to read "Despite the rhythmic behaviour of Tregs, they lack a functional, cell-autonomous clock, instead responding to systemic timing cues..."

11) Line 338: In the previous round of review, I had commented on the interpretation of the LN bioluminescence data. I don't think that the authors' revisions fully address my concern. I don't think that it is right to firmly conclude that T cells do not contribute to LN bioluminescence rhythms. It could be due to an incapacity of the assay to detect the reduction of amplitude in the KOs vs WT LNs. I would add the word "significantly" in this sentence: "... that T lymphocytes do not contribute significantly to the high amplitude rhythms..."

As suggested, we have added "significantly" to this sentence.

12) Lines 373-384: While the authors cite the papers from Shimba et al (ref 8) and Dimitrov et al (ref 48) in their discussion of the role of glucocorticoids in T cell function and trafficking, they do not mention that both of these papers showed that CXCR4 expression in T cells is regulated by glucocorticoids. This should be described.

We have now corrected this omission and have added the following sentence to the discussion "Indeed, earlier studies have demonstrated that application of glucocorticoids can induce CXCR4 expression on CD4⁺ T cells and CD8⁺ T cells^{8,47}."

13) The authors responded to a previous comment of mine, saying that "...we do not claim that this is the only (or main) mechanism driving diurnal regulation of Treg function. We agree that there are several other potential rhythmic signals that could contribute to the rhythmic phenotype we have observed, but it is beyond the limits of this manuscript to explore them here." I think that this should be mentioned in the Discussion.

As requested, we have now included the following sentence in our discussion to reflect our thoughts on how multiple rhythmic endogenous signals may contribute to the daily oscillations in the behaviour of Tregs in our arthritis model. "We present data that rhythmic glucocorticoid signals direct daily changes in the phenotype of Tregs, however we acknowledge that other rhythmic endogenous signals may also contribute to diurnal regulation of Treg function."

14) Lines 610-614: Description of ANOVAs here is fine, but I think that the p for the interaction and/or the main effects should be indicated in the figure legends. This was done by the authors in some cases, but not systematically.

We have now added this information into the manuscript for all relevant figures. Due to the complexity of the multi-panelled figures and the current length of the legends, we have added this information into a supplementary table (Supplementary Table 4).

Reviewer #2 (Remarks to the Author):

The authors have addressed my concerns.

REVIEWERS' COMMENTS:

Reviewer #1 (Remarks to the Author):

The authors have addressed all my concerns.